**Title**: An Autonomous Flow Through Salinity and Temperature Perturbation Mesocosm System
for Multi-stressor Experiments
**Author list**: Miller, C.A.[1,2*], Urrutti, P.[1], Gattuso, J.-P.[1,3], Comeau, S.[1], Lebrun, A.[1], Alliouane[1]
S., Schlegel, R.W.[1], and F. Gazeau[1]
[1]Sorbonne Université, CNRS, Laboratoire d'Océanographie de Villefranche, 181 chemin du
Lazaret, F-06230 Villefranche-sur-Mer, France
[2]Present address: Department of Earth Sciences, Geosciences, Utrecht University, Utrecht, The
Netherlands
[3]Institute for Sustainable Development and International Relations, Sciences Po, 27 rue Saint
Guillaume, F-75007 Paris, France
[*]*Correspondence to*: Cale A. Miller (e-mail: c.a.miller@uu.nl)

**Abstract**
The rapid environmental changes in aquatic systems as a result of anthropogenic forcings are
creating a multitude of challenging conditions for organisms and communities. The need to
better understand the interaction of environmental stressors now, and in the future, is
fundamental to determining the response of ecosystems to these perturbations. This work
describes an automated *ex-situ* mesocosm perturbation system that can manipulate several
variables of aquatic media in a controlled setting. This perturbation system was deployed in
Kongsfjorden (Svalbard) where ambient water from the fjord was heated and mixed with
freshwater in a multifactorial design to investigate the response of mixed kelp communities in
mesocosms to projected future Arctic conditions. The system employed an automated dynamic
offset scenario where a nominal temperature increase was programmed as a set value above real-
time ambient conditions in order to simulate future warming. A freshening component was
applied in a similar manner where a decrease in salinity was coupled to track the temperature
offset based on a temperature-salinity relationship in the fjord. The system functioned as an
automated mixing manifold that adjusted flow rates of warmed and chilled ambient seawater,
with unmanipulated ambient seawater and freshwater delivered as a single source of mixed
media to individual mesocosms. These conditions were maintained via continuously measured
temperature and salinity in all 12 mesocosms (1 control and 3 treatments, all in triplicates) for 54
days. System regulation was robust as median deviations from nominal conditions were < 0.15
for both temperature (°C) and salinity across the 3 replicates per treatment. Regulation further
improved during a second deployment that mimicked three marine heatwave scenarios where a
dynamic temperature regulation held median deviations to < 0.036°C from the nominal value for
all treatment conditions and replicates. This perturbation system has the potential to be
implemented across a wide range of conditions to test single or multi-stressor drivers (e.g.,
increased temperature, freshening, high $CO_2$) while maintaining natural variability. The
automated and independent control for each experimental unit (if desired) provides a large
breadth of versatility with respect to experimental design.

**1 Introduction**
The persistent burning of fossil fuels since the industrial revolution has radically increased
atmospheric $CO_2$. This has led to an enhanced greenhouse effect resulting in a multitude of
changing climatic elements such as increasing sea surface temperature (Bindoff et al., 2019). In
fjord systems, the confluence of increased fluvial inputs, glacier and permafrost meltwater,
stratification and water mass intrusion, as well as increased sea surface temperatures can create
periods of extreme physicochemical conditions for nearshore benthic and pelagic marine
communities (Bhatia et al., 2013; Poloczanska et al., 2016; Divya and Krishnan, 2017; Bindoff et
al., 2019). As ocean changes progress, the need to better understand the effects of combined
stressors (e.g., increased temperature and freshening) on marine communities is essential to
understand how community function and species richness will be affected while ecosystems
adjust to these new environmental conditions (Kroeker et al., 2017; Wake, 2019; Orr et al.,
2020). Several methodological approaches have been used to assess and characterize the
response of organisms and communities to future ocean changes, such as *ex-situ*
experimentation, the use of natural analogues (e.g., $CO_2$ vents), and space-for-time substitution
(using spatial phenomena to model temporal changes) (Blois et al., 2013; Rastrick et al., 2018;
Bass et al., 2021). These approaches, however, can be limited from testing the full range and
dynamics of present and future environmental conditions. The use of *ex-situ* experimental
systems that manipulate multiple environmental conditions, such as temperature and salinity, can
therefore provide a valuable tool to assess the response to multi-stressors in a future ocean.

The necessity of conducting multi-stressor experiments has become more pressing due to

the increasing interactions of environmental drivers within dynamic systems under a changing
climate (Kroeker et al., 2020). Nearshore regions can experience amplified modulations of
temperature and salinity on short timescales (Evans et al., 2015; Hales et al., 2016; Fairchild and
Hales, 2021). Such instances have been observed in sub-Arctic estuaries where water
temperature at a depth of 10 m decreased by 1.5°C in < 10 h, and in temperate systems where the
magnitude of salinity change driven by high precipitation displayed a decrease of 4 units in < 24
h (Miller and Kelley, 2021; Poppeschi et al., 2021). Changes of this magnitude are particularly
pertinent for Arctic fjords, where the variations in salinity from glacial meltwater can influence
whether a system exhibits net heterotrophic or autotrophic characteristics (Sejr et al., 2022).

Recent advances in the ability to modulate several environmental parameters at once

using *ex-situ* mesocosms have been made via the use of modular programmable systems (Wahl
et al., 2015; Pansch and Hiebenthal, 2019). Such systems have demonstrated an ability to apply
programmable environmental scenarios as a multifactorial design, or as a delta-change (offset)
from ambient conditions that mimic the natural variability of an environment. The advantages of
these types of automated systems lie in their ability to overcome the need for capturing and
measuring abundant discrete measurements used to regulate experimental conditions, and
transcend the logistical difficulties of implementing natural variability to experimental designs.
In addition, these systems can reduce the need for constant human observation which may be
required to program new regulatory operations or make rapid adjustments to experimentally
manipulated conditions.

Here, we describe an autonomous salinity and temperature experimental perturbation

mesocosm system (SalTExPreS) that has the ability to modify, and then regulate, salinity and
temperature in real-time. The SalTExPreS can perform similar functions as the *ex-situ* mesocosm
systems discussed above (i.e., Kiel-outdoor and -indoor benthocosms), such as applying
programmable static or dynamic changes to temperature and salinity, or by replicating natural
variability as on offset in real-time, but has the added capability of autonomous control for each
experimental unit (e.g., chamber or mesocosm). In the initial deployment of the SalTExPreS, we
applied a delta offset (i.e., offset from a measured control) to temperature and salinity as a
fractional-factorial treatment design for a two-month long experiment in KongsFjorden,
Svalbard, that exposed mixed kelp communities to future temperature, salinity, and irradiance.
This study demonstrates the stability and flexibility of the SalTExPreS as an experimental tool to
be utilized under extreme and dynamic conditions to test the effects of physicochemical multi-
stressors on marine organisms and communities in the context of a multi-month experiment.

**2 Methods**
**2.1 Operational Concept of the Experimental System**:
The SalTExPreS simulates the drivers in a marine or freshwater system such as temperature,
freshening, acidification, or hypoxia as either static or as temporally-variable modifications to a
reference water source. This is accomplished by mixing manipulated source water, whether it is
freshwater or warmed water, with ambient water through automatic flow valves that control the
volume and rate of water delivered. This is regulated by the constant monitoring of the mixed
water conditions in each mesocosm or chamber via a programmable feedback loop that transmits
the opening or closing of the automatic flow valves. The automated ability of the SalTExPreS is
configured to respond to near instantaneous measurements (several reads per second) to achieve
high frequency regulation of the manipulated drivers based on a measured *in-situ* or control
reference. The programmable nominal conditions in each mesocosm are easily controllable
through an intuitive user interface.

**2.2 Site Description and Experimental Design**
Kongsfjorden is a fjord system on the west coast of Svalbard (Norway) where the West
Spitsbergen Current exchanges warm Atlantic water through sill channels based on differences in
density gradients at the fjord mouth. Over the past two decades, a persistent influx of Atlantic
water has resulted in the reduction of sea ice and the melting of marine-terminating glaciers
causing enhanced freshwater and fluvial input (Luckman et al., 2015; Tverberg et al., 2019). The
influx of freshwater is highest in summer and is accompanied by an important sediment loading
with the potential to reduce the euphotic zone from 30 to 0.3 m depth (Svendsen et al., 2002).
These climatic changes in the Kongsfjorden environment set a relevant context for the inaugural
experiment of the SalTExPreS. It was placed on a concrete platform situated ~ 12 m from the
shoreline in Ny-Ålesund, which is located on southwestern shore of Kongsfjorden ~ 11 km from
the fjord mouth.

The SalTExPreS was utilized to implement three treatment scenarios in a fractional-

factorial design to simulate expected future conditions in Kongsfjorden for a 54-d experiment
that supervised the productivity, survival, and growth response of mixed kelp communities
surveyed at 7 m (maximum depth of collection). The treatments were realized by multi-driver
combinations of temperature, freshening, and irradiance, where treatments 1 and 2 differed in the
magnitude of temperature increase, salinity decrease, and irradiance decrease (Table 1). Only
temperature was manipulated for treatment 3. The chosen treatment and salinity perturbations
were applied as offset values from *in-situ* fjord conditions, which were measured at an
underwater observatory fixed at 11 m depth and captured the natural variability of the fjord
system. The applied temperature offsets used for this experiment reflected the projected SSP2-
4.5 and SSP5-8.5 scenarios (Meredith et al., 2019; Overland et al., 2019; Table 1). The chosen
decreases in salinity were based on correlations between *in-situ* temperature and salinity during
summer 2020 in Kongsfjorden (Gattuso et al., 2023), weeks 22 to 35 (Appendix A1 and Fig.
A1). These calculated delta salinity values were applied as offsets in treatments 1 and 2 (Table
1). The third treatment scenario applied a temperature change of + 5.3°C as a way to decouple
the multi-stressor system and evaluate a temperature only stress. The effect of turbidity for
treatments 1 and 2 were simulated as a decrease in surface irradiance (i.e., ~ 25% and ~ 40%
reduction from ambient irradiance at 7 m) by applying a combination of neutral light and spectral
filters (Lee© Filters) placed as static fixtures over the top of the mesocosms. The response of
these kelp community assemblages was determined in part by conducting weekly closed system
incubations and assessing the growth and metabolism of the kelp in each mesocosm—details and
results of this experiment are discussed elsewhere (Lebrun et al. *in review*; Miller et al., *in*
*review*).

**2.3 Experimental System**
Water was pumped from Kongsfjorden at a 10 m depth (300 m offshore) using a submersible
pump (NPS© Albatros F13T) that was tapped into an underwater intake pipe and that fed a
header tank in the Kings Bay Marine Laboratory in Ny-Ålesund, Svalbard. To prevent clogging
from sediment, the pump was situated at a 10 m depth ensuring a safe height above sediment
resuspension from the floor. Pumped ambient seawater from the header tank was then split into
three sub-header tanks within the marine lab where ambient water was (1) left unchanged, (2)
chilled to 0°C, or (3) warmed to 15°C. Each sub-header tank was plumbed to supply a maximum
flow of 6 $m^3$ $h^{-1}$ for the ambient, 1 $m^3$ $h^{-1}$ for chilled, and 2 $m^3$ $h^{-1}$ of warmed water which
required a pressure of 0.3 bars for each line to ensure consistent flow rates (Fig. 1). The three
control mesocosms received a mix of chilled and ambient seawater in order to properly simulate
*in-situ* temperatures. The three experimental treatments (nine mesocosms in total) received a mix
of ambient, warmed, and freshwater for treatments 1 and 2, whereas treatment 3 received a mix
of just ambient and warmed water (Fig 1). Freshwater was sourced from the tap which is fed by
the Tvillingvann reservoir close to Ny-Ålesund. The total flow-through rate of each mesocosm
was 0.5 $m^3$ $h^{-1}$ (i.e., each mesocosm turned over every 2 h) of post-mixed media delivered in an
open cycle flow-through system, which was the necessary flow rate needed to maintain the target
nominal values. Continuous flow was maintained throughout the experiment except for weekly 3
h interruptions (to perform experiments on the community) where the flow to each mesocosm
was shut off. In total, there were twelve circular mesocosms (3 treatments and 1 control, each
with 3 replicates) with a mean diameter of 1.1 m and a volume of 1 $m^3$, each equipped with a 12
W wave pump (Sunsun© JVP-132), a temperature-conductivity probe (Aqualabo, PC4E), an
optical oxygen sensor (Aqualabo, PODOC), and an Odyssey© light logger. Fiberglass insulation
at the outside of each mesocosm reduced unintended changes in treatment water temperature.

Delivery of ambient, chilled, warmed and freshwater first ran through an automated

mixing manifold that regulated the flow of each media type assuring that proper volumetric
proportions passed through the regulator valves to achieve target conditions (Fig. 1). Each
source-water flow line was regulated by an automated 2-way mixing valve (including the
incoming freshwater line) which then passed through a 3-way mixing valve that was assigned to
each mesocosm (12 in total, Fig. 1). This style of regulation ensured that the proper proportions
of manipulated media and ambient water were mixed to achieve nominal conditions. Any
temperature variation induced by mixing freshwater was immediately compensated for by
regulating the flow of the warm water line. Details regarding the programmed regulation are
discussed further in the appendix (Section A2). The mixed media then passed through a flow
meter which measured the flow rate to each mesocosm. A hand-crank regulating valve was
placed directly after the flow meter and was used for making minor adjustments and controlling
the overall flow. Measurements by the pressure sensors, the status of open position for the
regulator valves, and flow rates were logged every minute and displayed on the user interface
(Fig. A3).

## 2.4 Nominal Regulation

Nominal temperature conditions of + 3.3, 5.3, and 5.3°C applied to treatments 1, 2, and 3,
respectively, were offsets from the nominal control temperature. The nominal temperature of the
control was updated hourly and programmed to replicate the measured i*n-situ* conditions in the
fjord logged by the AWIPEV (Alfred Wegener Institute and Institute Paul Emile Victor)
FerryBox part of the COSYNA underwater observatory (https://dashboard.awi.de/) situated at a
depth of 11 m. Each treatment condition (temperature and salinity offset) was set by manually
programming the nominal value of temperature in the software interface (see appendix A3). The
salinity offset was coupled to the nominal temperature via the correlation described in appendix
A1. The measured temperature and salinity observations from inside each mesocosm were
recorded multiple times per minute and used to continuously monitor the regulation of the
conditions inside each mesocosm. This data transmission was used to program the software
controller that performed the automated regulation of mixed media (for details see appendix A2).
**2.5 Software**
The software application used for the control of the SalTExPreS was developed using Visual
Studio Community (2019 edition) with the vMicro extension and Arduino 1.8.13. The program
application has a user-friendly interface designed to allow real-time monitoring and
parameterization of regulation processes (Fig. A3). The main window displays each mesocosm
condition (the parameters measured by a sensor), their piping connections, a connection status
for each Programmable Logic Controller (PLC) informing on proper communication, date and
time of the last received communication packet from the Head PLC, and the status of the
experiment (e.g., started or stopped). The interface also displays the valve opening percentage
along with the nominal pressure and the actual measured value for each main source-water inlet.
In addition, the *in-situ* data (temperature and salinity) received from the FerryBox is displayed
with the time and date of the last logged value utilized to program the real-time nominal value of
the control. Sensor readings of flow rate (L min$^{-1}$), $O_2$ saturation (%), salinity, and temperature
(°C) are shown for each mesocosm in conjunction with the treatment nominal values (i.e.,
temperature, and salinity when relevant). All measured data are stored through the server
connection to the cloud, however, there is a backup microSD card on the Head PLC that logs
data from all mesocosms every 5 sec. If communication fails between the Head PLC and the
interfaced computer, data will not be retrieved by the PC during the communication break but
will be retained by the microSD card.
**3 Results**


**3.1 Regulation of the Control**



The control was able to simulate the ambient fjord temperature well over the



experimental period where the average value across the 3 replicates deviated < 0.3°C (Table 2,


Fig. 2). The overall quality of the regulation was achieved by the ability of the system to


interpret and respond to the measured data from the FerryBox (or to follow a manually


programmed nominal value when communication with the FerryBox was interrupted). During


the experiment, the FerryBox went intermittently offline 24% of the time, ceasing transmission


of real-time data that resulted in a break of communication to the PLCs. This somewhat frequent


break in communication resulted in an average nominal deviation that was nearly double for the


control compared to the treatment conditions (Table 2). The ability to manually program a new


nominal value when communication breaks occurred ensured that the control remained robustly


regulated. Over the entire period of the SalTExPreS deployment, the mean temperature of the


control increased from ~ 4 to 6.5°C from early July to the end of August (Fig. 3a). The coldest


mean temperature of the control occurred when a backup pump situated at 90 m depth in the


fjord was used from 2021-07-14 ~21:00 UTC until 2021-07-26 13:49 UTC while the original


pump at 10 m depth was repaired. During this period, the control was ~ 1.0 – 1.5°C cooler than


the temperature measured by the FerryBox (Figs. 2, 3). Since a warmed seawater inlet was not


supplied to the control, the temperature of the control remained cooler than the measured


ambient conditions at the FerryBox. Despite the cooler temperature for the control, regulation of


flow rates, mesocosm turnover time, and variability across the control replicates was well


maintained by the system.



**1.2 Temperature and Salinity Regulation**



The regulation of temperature and salinity in the different treatment conditions (Trts. 1 – 3) was
maintained by the SalTExPreS for the full planned duration of 54 days (2021-07-03 to 2021-08-
26). For the first 6 days of the SalTExPreS experiment, the treatment conditions were held at the
control (i.e., no applied offset from the control) before the stepwise increase in temperature
began. On 2021-07-10 12:00 UTC a temperature offset of 0.55°C $d^{-1}$ was programmed for
treatment 1 while treatment 2 and 3 were programmed to increase by 0.88°C $d^{-1}$ (Figs. 2, 3). The
final nominal temperature above the control was reached on 2021-07-15 21:00 UTC. The system
needed 4 h to achieve the new temperature conditions (i.e., homogenize the mesocosm to a
0.88°C increase). A manual override was applied to the salinity regulation for treatments 1 and 2
which resulted in the system achieving the final salinity offset value upon the initial temperature
increase (Fig. 3b, 4). This was done to ensure the maintenance of salinity regulation as the
temperature offsets were applied relative to the control, which was receiving fjord water pumped
from 90 m and was colder than the measured *in-situ* conditions. It took the system 4 h to achieve
the salinity offset for treatment 2 adjusting the value from ~ 34 to 29.8 (Fig. 3b, 4).
The precision of the temperature and salinity regulation across all treatment conditions
was well maintained as the mean difference between the measured value and the nominal value
was < 0.2°C and < 0.36 for salinity across the entire deployment (Table 2). The mean deviations
observed across treatments did not appear to correlate to the degree of offset. Thus, treatment 3
showed the highest precision for temperature regulation, while salinity regulation was the most
robust for treatment 2 compared to treatment 1 (Table 2). During several instances when
communication was interrupted between the FerryBox and the Head PLC, the SalTExPreS
retained the last measured value at the FerryBox as a contingency protocol. This aided in the
ability of the system to maintain a high degree of regulation throughout the entire deployment.
The largest deviation from the nominal value for all treatment conditions occurred during the
single instance in which the last read value from the FerryBox was not retained: this occurred on
2021-08-24 04:47 UTC (Fig. 4). Communication was quickly restored after this incident by
cycling the program code, and the average deviation of temperature (°C) and salinity for
treatment 1 for the remainder of the deployment was < 0.16, and < 0.25 for treatment 2.

When adequate flow rates were maintained, the SalTExPreS was able to simultaneously

regulate 12 mesocosms at 4 different conditions to deviations in temperature and salinity that
were < 0.5°C or 0.5 in salinity from the nominal value ≥ 80% and ≥ 70% of the time,
respectively (Fig. 5). Due to an erroneous nominal value for the control during the 90 m pump
usage, these times were excluded. If warm water could have been mixed with the ambient water
feeding the control mesocosms, then a proper nominal value could have been maintained. Over
the full duration of the experiment, effective regulation from the nominal temperature and
salinity values were kept  to < 1 for all mesocosms 89% of the time for temperature (°C), and
80% for the salinity (excluding the 1[st] replicate for treatment 2).

**Discussion**

The first application of the fully autonomous SalTExPreS demonstrated the capacity of the
system to successfully manipulate temperature and salinity as an offset value from the control,
thus maintaining, natural, *in-situ* variability for 4 different conditions simultaneously. We
utilized this deployment to test the effects of climate change drivers on Arctic kelp communities
recognizing the feasibility of the system to perform *ex-situ* experiments on organisms or whole
communities (Miller et al., *in review*). The versatility of the system not only allows for the
manipulation of temperature and salinity, but can incorporate other factors such as $CO_2$ or
hypoxia (Gazeau et al., *in prep*). While this experiment used a control offset approach to produce
treatment conditions, programmable parametrization of various treatment combinations can be
applied depending on the question and design of the experiment. The automated component of
the system reduced the logistical hurdles that can arise when performing high precision
replication and regulation of experimental conditions that track real-time system variability.
While the use of such a system can reduce user oversight and limitations, there is still a need for
diligent operation.

Since the initial experiment, we have implemented a number of changes to improve the

performance of the system which have been realized during a second experiment in the summer
of 2022 (Fig. 6). In this experiment, the SalTExPreS was integrated to function with a deployable
heat pump to simulate multiple scenarios of marine heatwave patterns over a nearly month-long
experiment. In this instance, temperature regulation was vastly improved as a result of the
programmable modifications made since the initial experiment. During this second experiment,
the SalTExPreS mimicked 3 marine heatwave scenarios where a dynamic temperature regulation
kept deviations in the 9 different mesocosms at < 0.5°C for 94% of the time. This was an
improvement to the % time of temperature regulation by ∼ 15% compared to the first
experiment. During the first experiment, inconsistent flow rates and communication errors
between the FerryBox and the Head PLC were the primary causes of larger deviations (> 2.0
salinity or °C) from nominal values. For example, flow rates of < 2 L min$^{-1}$ accounted for ∼ 20%
of the large deviations in temperature and salinity regulation. Simple software modifications
such as 'pop-up' alert windows that warned when a lapse in communication with the FerryBox
occurred (e.g., FerryBox stopped logging), and the addition of contingency coding instructions
(i.e., fail-safe instructions) ensuring that the last received *in-situ* data were maintained solved
most of the issues. Communication errors were easily remedied by cycling the power on a PLC,
which is why pop-up alerts were an improvement to the operation. Other extraneous
circumstances that could impact flow rates, such as pump failure and clogging of the seawater
intake ports, are issues that need to be addressed whenever the SalTExPreS is used. However,
these are very manageable situations which can be easily mitigated by an operator.
The novelty of the SalTExPreS lies in its ability to independently regulate experimental
conditions in a single experimental chamber (e.g., mesocosm). The operational data produced
from this deployment are reliable, easily quantifiable, and provide the highest degree of
monitoring frequency for every applied experimental condition. This study has demonstrated the
system's ability to replicate dynamic nearshore environments where temperature and salinity can
vary at high frequency (e.g., tidally). The system's additional capacity to mimic future scenarios
by applying an amplitude offset to the natural dynamics of *in-situ* conditions is an added feature
for conducting manipulative experiments. Wahl et al. (2015) described a system with a similar
capability, but regulated treatment conditions by monitoring source water and adjusting that
media before it was delivered to each experimental chamber. The SalTExPreS differs in that it
measures the conditions inside each experimental chamber (i.e., mesocosm) and regulates them
independently based on per second measurements. This provides the flexibility to individually
modulate each experimental chamber providing a broad range of versatility. The lack of
infrastructure needed to set up the SalTExPreS makes it easy to deploy and transport. As long as
there is a sufficient supply of ambient water and manipulated media, there is little limit to the
versatility of automated control for each mesocosm. Many research endeavors and future
implementations by the SalTExPreS have the potential to conduct a large range of experimental
settings that pertain to environmental perturbations associated with climate change or other

anthropogenic forcings. The operation of such a system in extreme environmental conditions has

shown the durability of the manifold to endure an adverse Arctic summer and still respond

without mechanical failures. With proper operation and user proficiency, this proves to be a

highly sophisticated and powerful tool to be utilized for marine and aquatic perturbation

experiments.

**Acknowledgements**

This study is part of the FACE-IT Project (The Future of Arctic Coastal Ecosystems –

Identifying Transitions in Fjord Systems and Adjacent Coastal Areas). The authors thank Jens

Terhaar for helping with temperature projection data, Philipp Fischer for access to the AWIPEV

data as well as AWIPEV and Kings Bay staff for helping with logistical details, shipping, and

access to the marine lab facilities.

**Author contributions**

C.M. and F.G. conceptualized the frame of the paper while F.G, S.C, and P.U. designed the

experimental system. P.U. programmed the software. C.M. wrote the manuscript, performed the

data analysis, and constructed the figures and tables while P.U. designed schematic figures. All

authors participated in the operation of the system and have, thus, commented, and edited during

writing.

**Financial support**

This study was conducted in the frame of the project FACE-IT (The Future of Arctic Coastal

Ecosystems – Identifying Transitions in Fjord Systems and Adjacent Coastal Areas). FACE-IT

has received funding from the European Union's Horizon 2020 research and innovation
programme under grant agreement No 869154. Logistical and financial support was provided by
IPEV, The French Polar Institute and the Foundation Prince Albert 2 of Monaco (project: 3051,
http/fpa2.org).

**Competing interest**
The authors declare no competing interests exist.

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

**Tables**
**Table 1.** Experimental treatment conditions with corresponding offsets (as compared to the
control) for temperature (°C), salinity and photosynthetically active radiation (PAR; expressed as
a percentage). See section A1 and figure A1 for a full description of the temperature-salinity
relationship used to calculate salinity offsets.

| Treatment | Temperature | Salinity | PAR |
|:---:|:---:|:---:|:---:|
| 1 | + 3.3 °C | - 2.5 – 3.0<br>- S = 0.546*T + 0.490 | - 25% PAR |
| 2 | + 5.3 °C | - 5.0 – 5.5<br>- S = 0.877*T + 0.089 | - 40% PAR |
| 3 | + 5.3 °C | Ambient | Ambient |















**Table 2.** Absolute mean difference between measured temperature ($T_{meas}$; °C) and salinity ($S_{meas}$)
values against nominal values ($T_{nominal}$ and $S_{nominal}$) plus or minus the corresponding standard
deviation, in each mesocosm during the experimental period. A weighted average was used for
treatments 1 – 3 to account for the initial 5-day incremental increase. Triplicate mesocosms per
condition are expressed as a, b and c. Water mixture indicates the types of media supplied to
each treatment, denoted with an 'x'.

| Treatment | Mean diff $Abs(T_{meas.} - T_{nominal})$ | Mean diff $Abs(S_{meas.} - S_{nominal})$ | Water mixture | | | |
|---|---|---|---|---|---|---|
| | | | Cold | Ambient | Warm | Fresh |
| Control a | 0.275 ± 0.39 | – | x | x | | |
| Control b | 0.291 ± 0.36 | – | x | x | | |
| Control c | 0.223 ± 0.36 | – | x | x | | |
| Treatment 1a | 0.126 ± 0.31 | 0.116 ± 0.31 | | x | x | x |
| Treatment 1b | 0.142 ± 0.29 | 0.148 ± 0.22 | | x | x | x |
| Treatment 1c | 0.145 ± 0.33 | 0.171 ± 0.33 | | x | x | x |
| Treatment 2a | 0.111 ± 0.29 | 0.357 ± 0.74 | | x | x | x |
| Treatment 2b | 0.133 ± 0.29 | 0.149 ± 0.26 | | x | x | x |
| Treatment 2c | 0.196 ± 0.38 | 0.128 ± 0.25 | | x | x | x |
| Treatment 3a | 0.109 ± 0.27 | – | | x | x | |
| Treatment 3b | 0.112 ± 0.27 | – | | x | x | |
| Treatment 3c | 0.106 ± 0.28 | – | | x | x | |













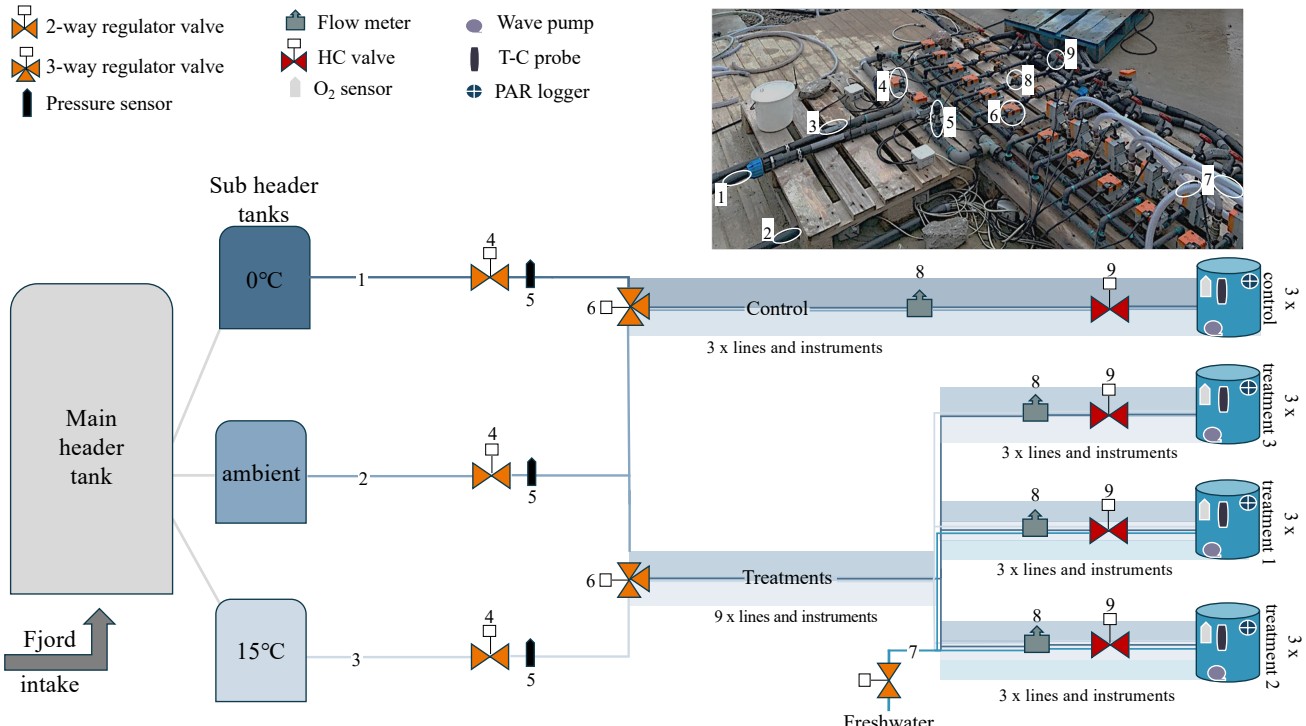


**Figure 1.** Piping schematic of the SalTExPreS which includes the mixing and regulation

manifold. Items 1 – 3 depict the main seawater inlets from the chilled, ambient, and warmed sub-

header tanks located in the Kings Bay Marine Laboratory. Seawater from each sub-header tank

moves through a 2-way regulator (4) valve followed by a pressure sensor (5) before splitting into

individual lines that lead to all 12 3-way regulator valves (6), each assigned to a single

mesocosm. For treatments 1 and 2, the freshwater inlet (clear tube; item 7) passes through a 2-

way regulator valve before mixing with the ambient and warmed seawater lines. Flow rates are

then measured (8) post-mixing, and final flow rates are set using a hand-crank (HC) red valve

(9). The shaded regions in the schematic indicate that mixed media lines and instruments occur

3x or 9x times. T-C probe is the temperature-conductivity probe and the PAR logger measures

the photosynthetically active radiation. Photos of mesocosms and the sensors inside can be found

in the appendix (Fig. A6). Table A1 provides the parts list for the items shown in this figure.

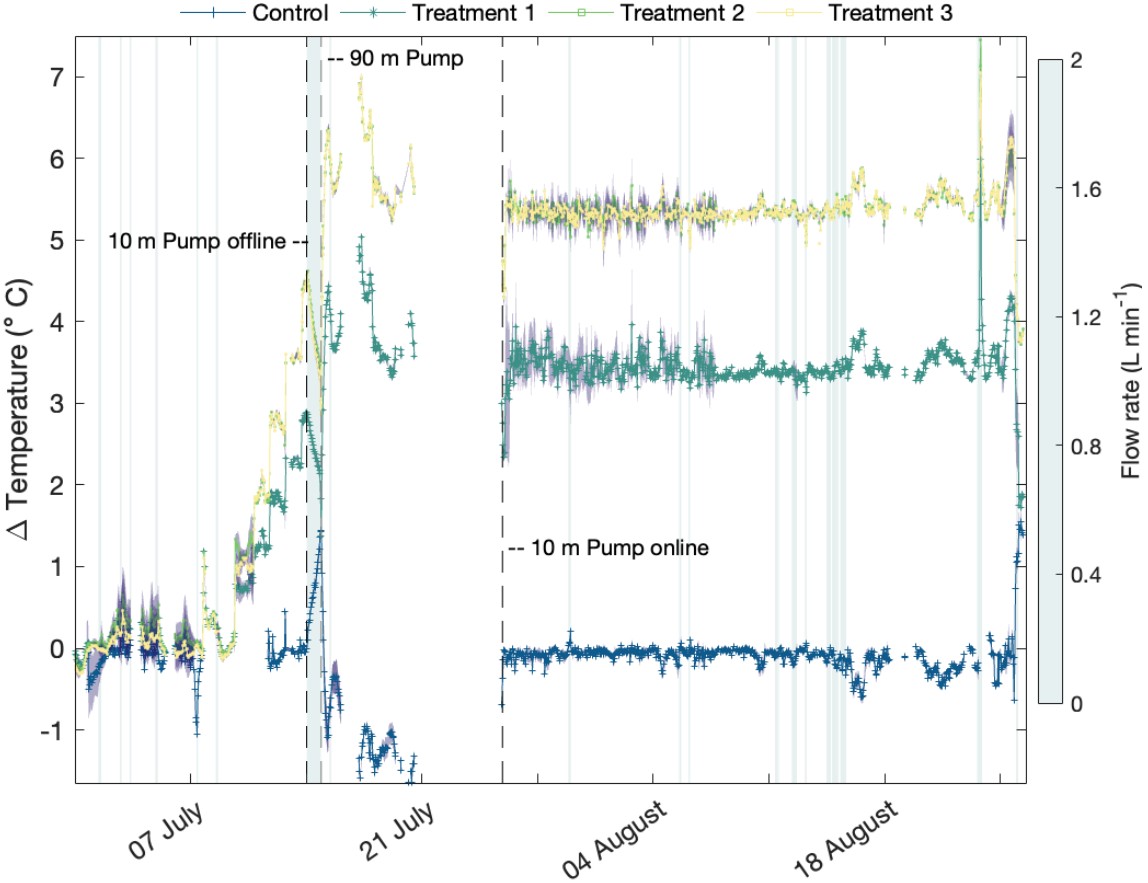

**Figure 2.** The hourly mean (across triplicated mesocosms) temperature offsets of all applied

conditions. For control mesocosms (in blue), offsets were calculated against *in-situ*

measurements (FerryBox). For the three experimental treatments (dark green, light green, and

yellow for treatments 1, 2 and 3, respectively), offsets were estimated against the mean control

values. The purple shaded region around the mean is the standard deviation. The heatmap

isoclines (blue-grey shaded regions) are instances when flow rates were $\leq 2$ L min$^{-1}$ (threshold to

avoid large deviations $> 2.0$ salinity or °C). Dashed black lines indicate periods when the pump

at 10 m depth and 90 m depth were used to feed the sub-header tanks. The time presented is the

duration of the experimental deployment.



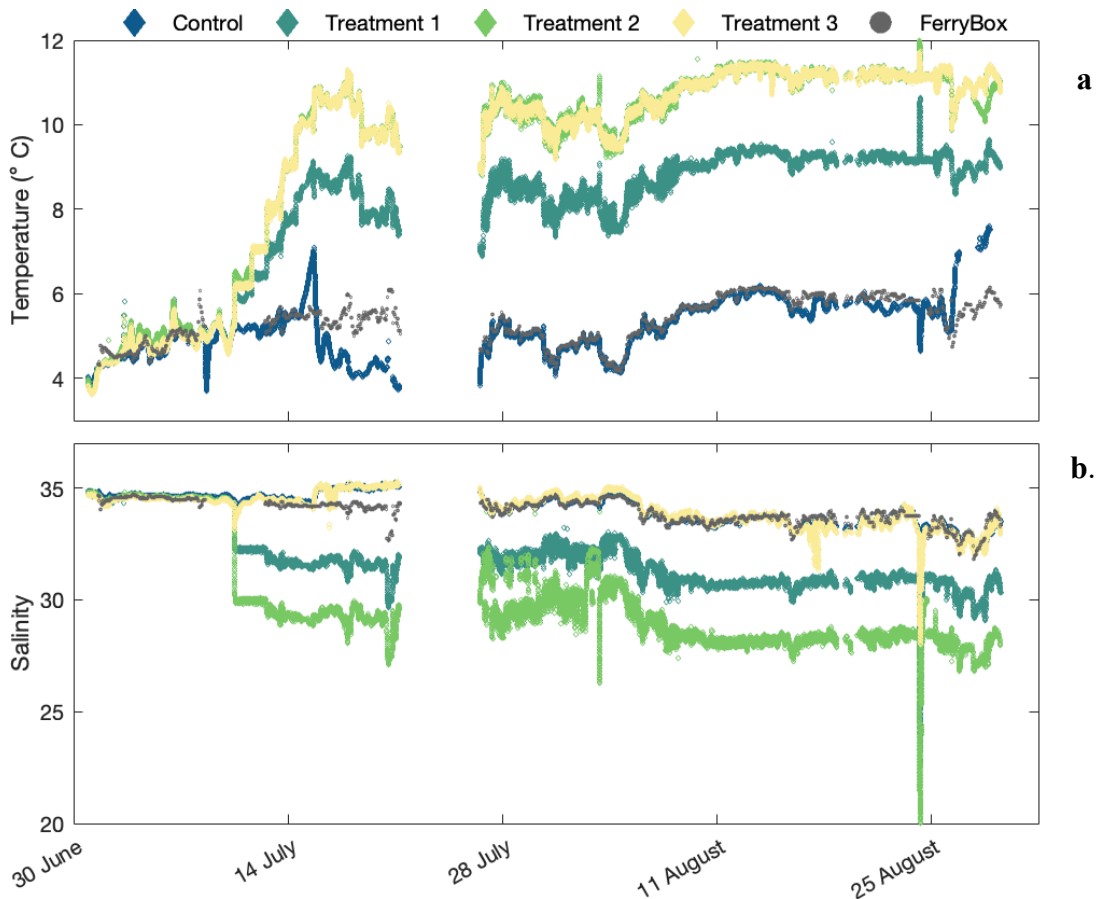


**Figure 3**. Mean (across triplicated mesocosms) temperature (°C; **a**) and salinity (**b**) values
measured every minute over a 60 d period (including 6 day period before the start of the
experiment) for the control (blue), and for treatments 1 – 3 (dark green, light green, and yellow,
respectively).




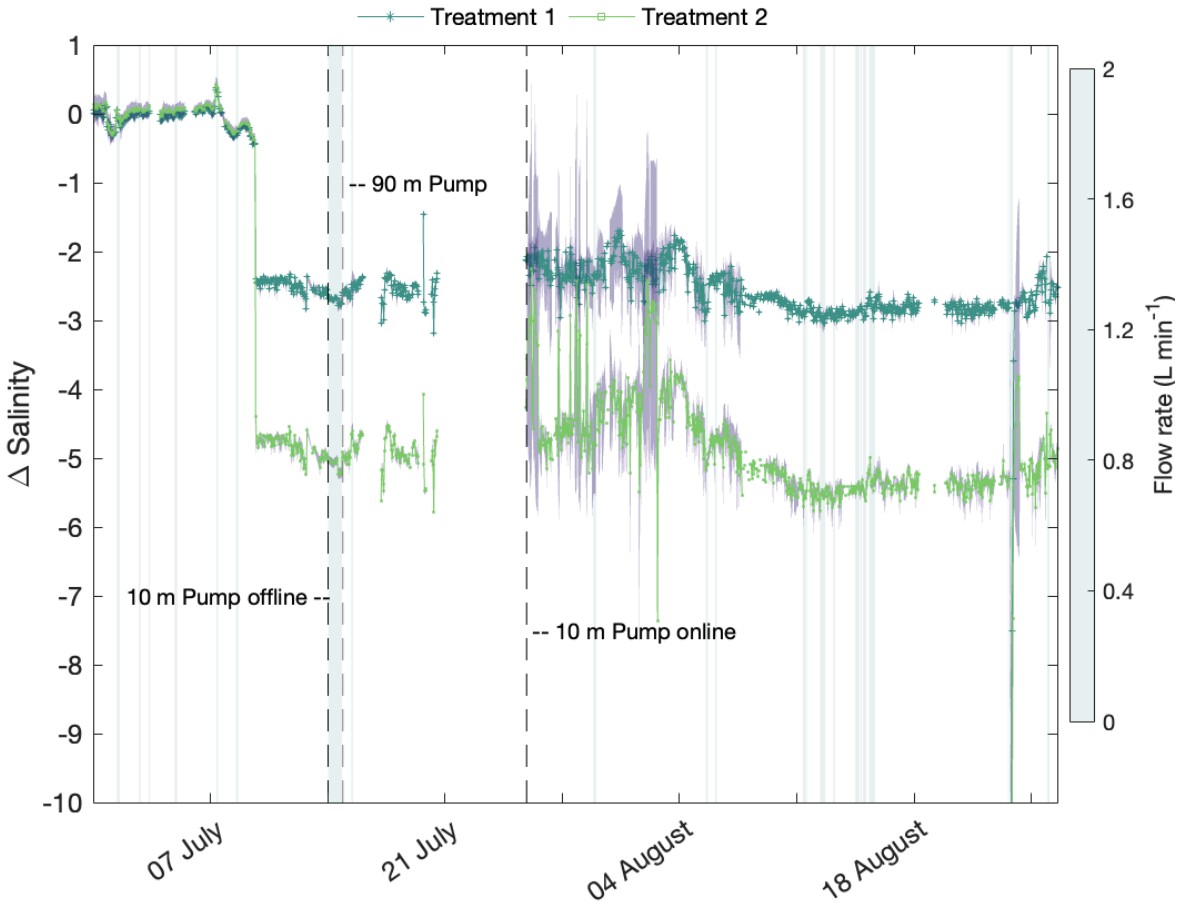


**Figure 4.** The hourly mean (across triplicated mesocosms) salinity offsets for the experimental

period. Dark green is treatment 1 and light green is treatment 2.The purple shaded region around

the mean is the standard deviation and the heatmap isoclines (blue shaded regions) are the

instances when flow rates $\leq 2$ L min$^{-1}$. Dashed black lines indicate periods when the pump at 10

m depth and 90 m depth were used to feed the sub-header tanks.







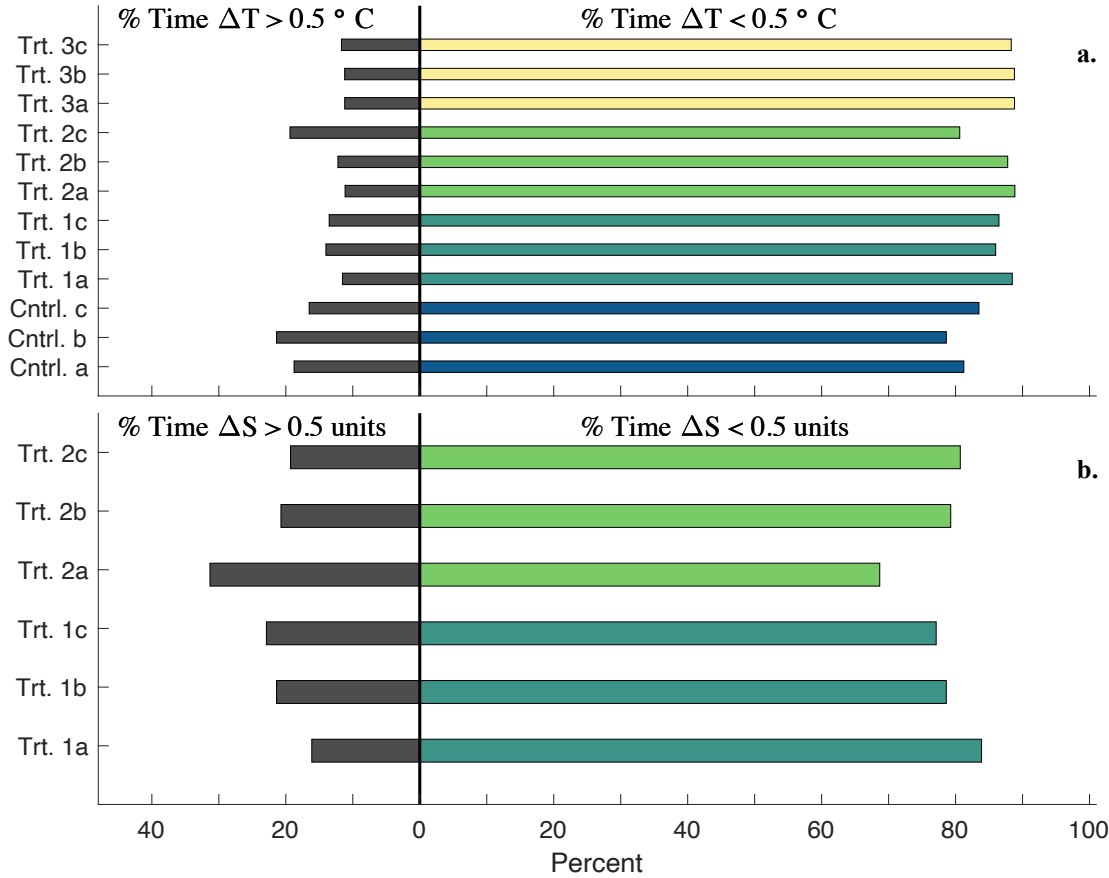


**Figure 5.** Percent time each mesocosm experienced a deviation > (black bars) or < (colored

bars) 0.5°C ($\Delta$T; **a**) or 0.5 in salinity ($\Delta$S; **b**) when flow rates were above 2 L min$^{-1}$. Cntrl. and

Trt. abbreviations are the control and treatments, respectively. This excludes the period when

using the 90 m pump (12 d), but accounts for 42 days out of the 54-day experiment. Bar color

indicates different treatment groups, as shown on the y-axes.







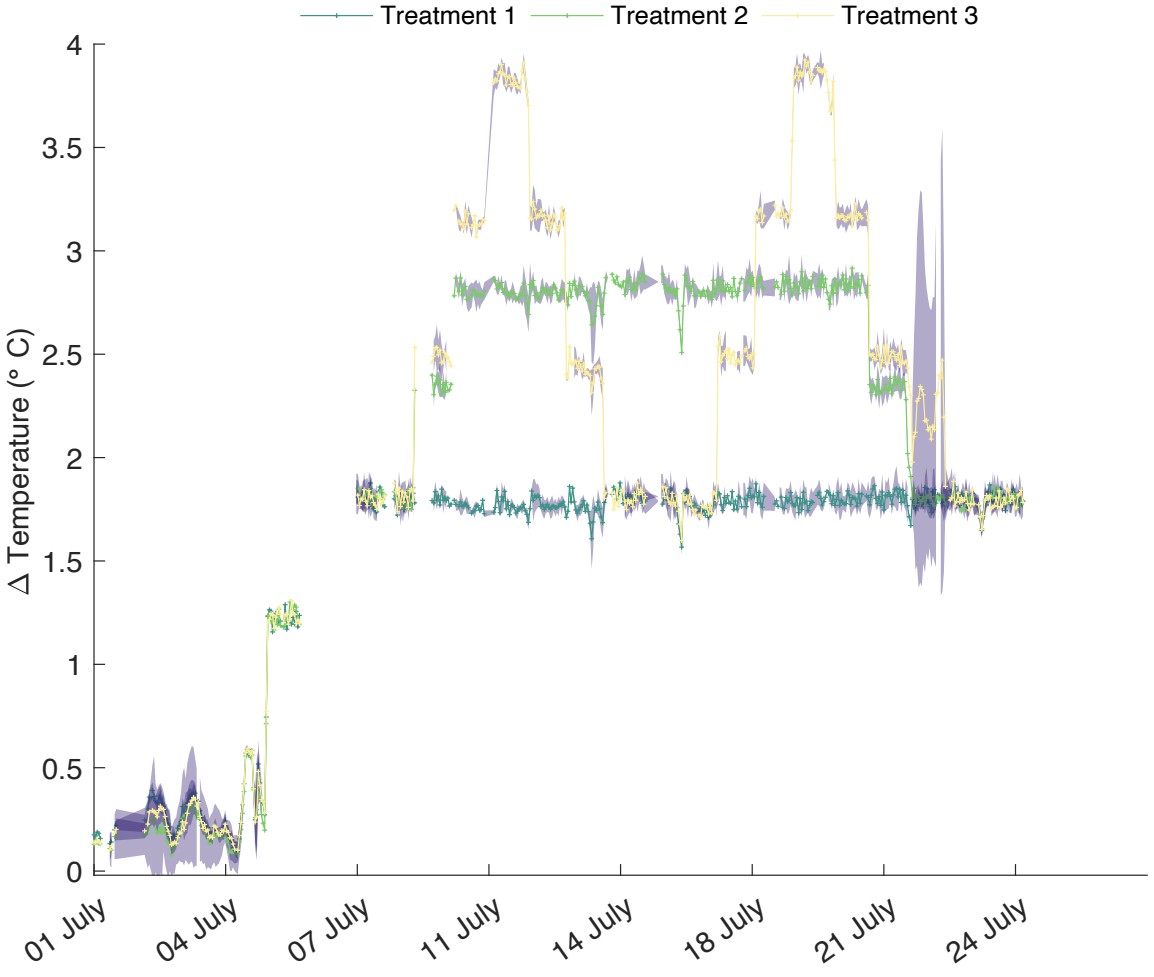


**Figure 6.** The hourly mean temperature offsets (Δ Temperature) during the 2$^{nd}$ deployment of

SalTExPreS in the summer of 2022 in Tromsø (Norway) performing a variation of heatwave

scenarios with three experimental treatments 1 – 3. Treatment 1 is a constant high temperature (+

1.76°C), treatment 2 is a low frequency (1 heatwave) and medium magnitude offset (+ 2.81°C),

while treatment 3 is a high frequency (2 heatwaves) and magnitude offset (+ 3.86°C). The purple

shaded region around the mean is the standard deviation.

**Appendix**
**A1. Calculation of Salinity Offset**
In the summer of 2020—weeks 22 to 35— the mean temperature at 11 m displayed a range from
2.48 – 6.28, with salinity values ranging from 34.67 measured at the minimum 2.48°C and 33.63
measured at 6.28°C (Fig. A1a). The correlation was best fit with a $2^{nd}$ order polynomial. To
project the salinity offset at a future temperature based on this $2^{nd}$ order polynomial fit,
temperatures of + 3.3 and 5.3°C (SSP2-4.5 and SSP5-8.5, respectively) were added to *in-situ*
fjord temperatures and salinity was calculated based on the $2^{nd}$ order polynomial. These
estimated salinity values were then subtracted from the mean salinity values observed (y-axis,
Fig. A1a) in summer 2020 in order to calculate a delta salinity value for the SSP2-4.5 and SSP5-
8.5 scenarios. The relationship between these estimated delta salinity values and the mean *in-situ*
temperature (y-axis, Fig. A1a) displayed a robust linear relationship (Fig A1b).

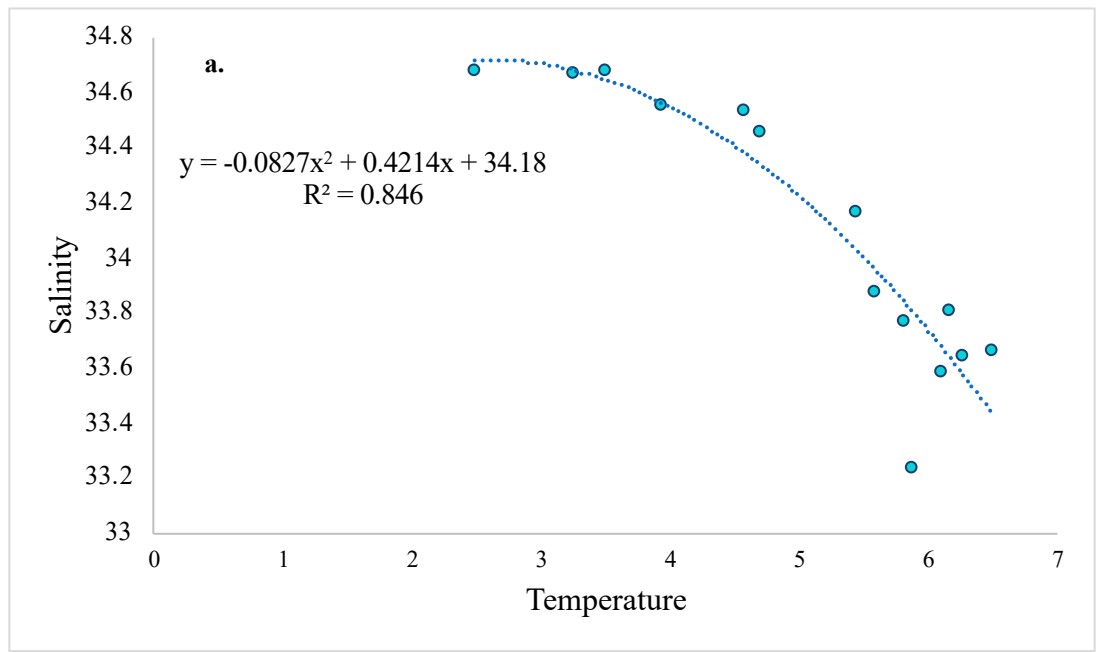


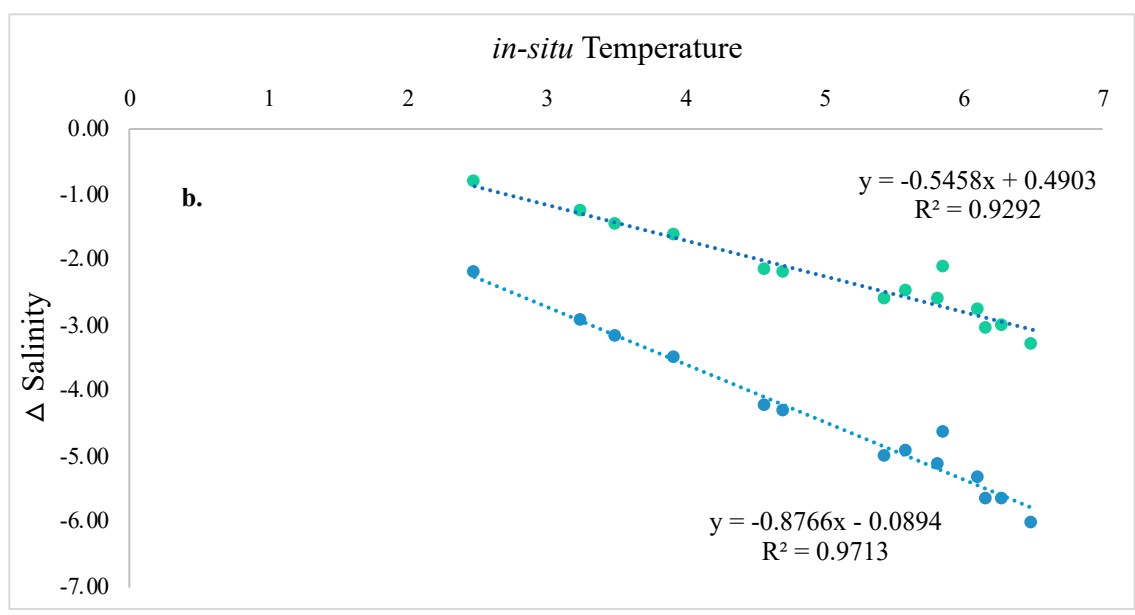


**Figure A1.** Relationship between temperature and salinity in summer 2020 weeks 22 – 35 in Ny-

Ålesund, Svalbard (**a**). Relationship between estimated delta salinity and *in-situ* temperature,

where delta salinity was calculated as the difference between the current mean salinity and the

salinity estimated at the temperature increase projected for SSP2-4.5 (green dots) and SSP5-8.5

(blue dots) scenarios (**b**).











**Table A1. Parts list with manufacturer model numbers.**

| Group | Item | Supplier/manufacturer | Model / details | Quantity |
|---|---|---|---|---|
| Hydraulic system | | | | |
| | Mesocosms | home made | 1000 L fiber glass | 12 |
| | Seawater pump | NPS, BradFord, UK | Albatros F13T | 1 |
| | PVC-U tubing and fittings | | 20mm, 32mm & 50mm diameter | _ |
| | Insulated flexible hose | | 19mm diameter | 100 m |
| Sensors | | | | |
| | Conductivity / temperature | Aqualabo, Champigny sur Marne, France | PC4E | 12 |
| | Oxygen | Aqualabo, Champigny sur Marne, France | PODOC | 12 |
| | Pressure | Siemens, Munich, Germany | 7MF1567-3BE00-1AA1 | 3 |
| | Flow rate | IFM, Essen, Germany | SV3150 | 12 |
| Actuators | | | | |
| | Pressure regulation valves | BELIMO, Hinwil, Switzerland | R2025-10-S2 with LR24A-SR motor | 3 |
| | Temperature regulation valves | BELIMO, Hinwil, Switzerland | R3015-10-S2 with LR24A-SR motor | 12 |
| | Salinity regulation valves | BELIMO, Hinwil, Switzerland | R2015-10-S2 with LR24A-SR motor | 6 |
| Automation cabinet | | | | |
| | Cabinet | Fibox, Espoo, Finland | FIB8120017N | 1 |
| | Security switch | KRAUS-NAIMER, Karlsruhe, germany | KNA002245 | 1 |
| | 12 vdc power supply | TDK Lambda, New York, USA | LAMDRL30-12-1 | 1 |
| | 24vdc power supply | TDK Lambda, New York, USA | LAMDRB240-24-1 | 1 |
| | PLC | Industrial Shields, Barcelona, Spain | Mduino-42+ | 4 |
| | Ethernet switch | HIRSCHMANN-INET, Neckartenzlingen, Germany | HIR942132002 | 1 |


## A2. Temperature and Salinity Regulation

Accurate temperature and salinity regulation was managed using the software PID (proportional integral derivative) controller on the corresponding Programmable Logic Controller (PLC). The PLC operated in PoE mode (power over ethernet) which builds a local area network (LAN) enabling use of Ethernet data cables to carry electrical power. The PID controller measures the difference between the measured value and the nominal value (i.e., the error). This calculates the position and adjustment of the valve opening by multiplying the error, the integral of this error, and the derivative of the error over time, by previously determined coefficients $K_p$ (proportional gain), $K_i$ (integral gain) and $K_d$ (derivative gain), respectively. These coefficients were obtained experimentally using the empirical method of Ziegler & Nichols (1943). These coefficient values may differ from one condition to another.

## A2.1. Pressure and Flow Regulation

Each sub-header tank inlet line of ambient, chilled and warmed seawater had its own pressure regulation system enabling equivalent pressure levels to be maintained. This regulation process aided in the ability to adjust flow rates for all mesocosms by using the hand-crank valves (Fig. 1). The system consisted of an analog pressure sensor (Siemens© 7MF1567-3BE00-1AA1) and a two-way analog valve (BELIMO© R2025-10-S2 with LR24A-SR motor). The pressure sensors were placed in-line directly after water from each sub-header tank passed through a regulator valve. The sensor ensured that pressure for each line was maintained at 0.3 bars by transmitting data to the system which then regulated the valve opening position of the incoming flow. A nominal pressure for all three sensors was predetermined during flow rate test trials. This

process took place during the setup of the system where the valve opening was adjusted using a
PID regulator (see A2) to maintain the defined nominal pressure.
**A2.2 Automation**
The automation was performed using 4 Industrial Arduino-based PLCs (Industrial shields©
Mduino-42+), with an individual PLC regulating the control and each treatment 1 – 3,
respectively. Each PLC was responsible for logging data and regulating a specific experimental
condition. The PLC regulating the control—identified as the Head PLC—was the primary device
responsible for communication with the branched PLCs and the monitoring computer (Fig. A2).
All monitoring was performed on a PC Windows application (Section A3) and responsible for:
(1) reading data received from the PLCs, (2) reading *in-situ* data received from the internet, (3)
displaying live data, (4) logging data and sending it to an FTP server, and (5) sending settings
and commands to the PLCs. Communication between the PLCs and the PC was ensured using
http WebSocket protocol on RJ45 ethernet cables. The communication between the PLCs and the
conductivity-temperature and oxygen sensors, flow rate sensors, and regulation valves was
executed using a half duplex RS485 (2 wires) protocol, with an analog 4-20mA and an analog 0-
10V signal, respectively. All PLCs and wired communication lines were housed in an electrical
box installed to an IP68 Fibox enclosure with a 400 V (3P+N+E) 32 A security switch (Fig. A6).
All the automation elements use low tension (12 Vdc or 24 Vdc) through circuit breakers and
fuses. The electrical box was protected with a 220 V socket.

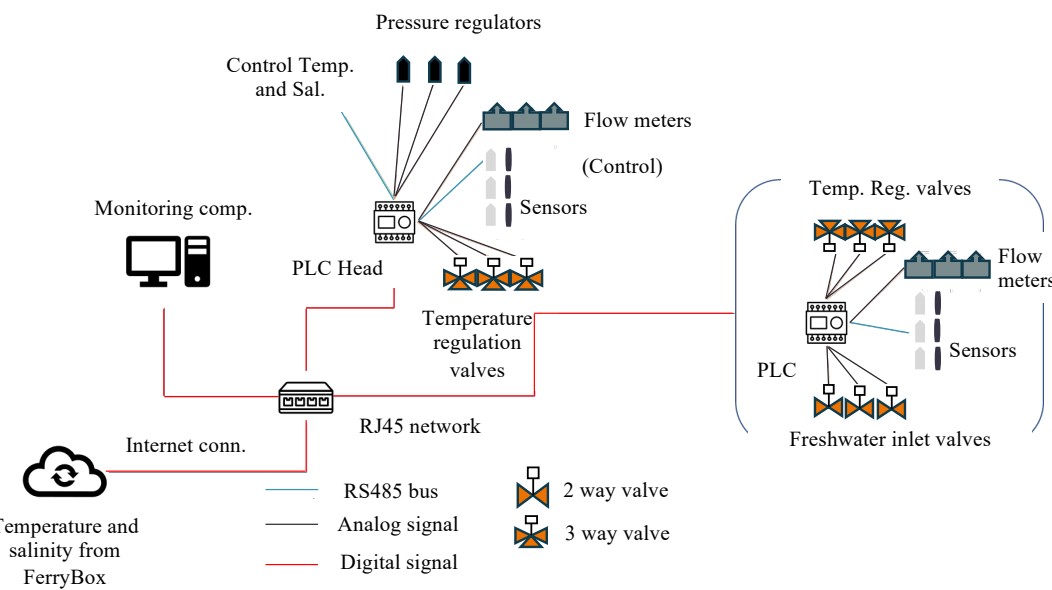


**Figure A2.** Diagram and flow-chart of the automation system.













**A3. Software Development**

The code for the application was written in C/C++. The code uses publicly available Arduino

libraries (https://www.arduino.cc/reference/en/libraries/) as well as originally designed libraries.

All code is available on Github (https://github.com/purrutti/FACEIT). The code is divided into

two pathways: 'Master.ino' for the Head PLC, and 'Regul_condition.ino' for the Branched

PLCs. A description of the main functions applied in the code for programming the system

regulation and features are listed in Table A3.

**Table A3.** Functions used for programming of software.

| Function | Operation | Ancillary field Sender ID | Ancillary field Command # |
|---|---|---|---|
| RTC.read() | The PLCs are equipped with a RTC chip and battery to keep track of the date. Once set on commissioning, RTC.read() returns the current date and time. | | |
| readMBSensors() | This functions loops through each sensor connected on the RS485 bus. Each Mesocosm has two sensors (O2 and Conductivity/Salinity), so each PLC has 6 sensors connected on its bus.<br>- O2 sensors have addresses ranging from 10 to 12, for mesocosms 0 to 2 of the scenario, respectively.<br>- PC4E sensors have addresses ranging from 30 to 32, for mesocosms 0 to 2 of the scenario, respectively.<br>- Sensors are requested individually and in sequence. A request is made every 200 ms. | | |
| webSocket.loop() | This is a callback function responsible for dealing with the WebSocket communication. The master PLC is the WebSocket server. It listens to slave PLCs requests and to the monitoring PC requests. Requests are JSON formatted. They always contain *ancillary fields*: senderID (ID of the entity sending the request), condID (ID of the requested entity), command (command type of the request). They optionally can also contain a « time » field: Unix-like timestamp (number of seconds since 01-01-1970) | Head PLC (ID = 0)<br>Branched PLCs (ID = 1–3)<br>Monitoring PC (ID = 4) | Request params: setpoints, PID settings (# = 0)<br>Request data: measurement values, regulation outputs (# = 1)<br>Send Params: response to a « request params » request (# = 2)<br>Send Data: response to a « request data » request (# = 3)<br>Calibrate sensor: request for calibrating sensor to specified value (# = 4)<br>Request Head data: specific data measured by Head PLC (pressure & flowrates ) (# = 5)<br>Send Head data: a response to a « request Head data » request (# = 6) |
| checkMesocosmes() | This functions loops through every mesocosm every 200 ms and reads analog signals (i.e., flowrates and pressure readings). | | |
| regulationTemperature() | This function is responsible for the temperature regulation of the mesocosm. It sets the corresponding three-way valve position using a 0-10V analog signal. The function first checks if the regulation is in « manual override » mode. If so, it applies the override setpoint. If not, it reads the temperature measure in the mesocosm, compares it with the setpoint, and uses the PID settings to set the valve position. | | |
| regulationPression()<br>*Only for HEAD PLC* | This function is responsible for the pressure regulation of the mesocosm. It sets the corresponding three-way valve position using a 0-10V analog signal. The function first checks if the regulation is in « manual override » mode. If so, it applies the override setpoint. If not, it reads the pressure measure in the mesocosm, compares it with the setpoint, and uses the PID settings to set the valve position. | | |
| printToSD()<br>*Only for HEAD PLC* | Master PLC is equipped with a microSD card, on which data from all mesocosms is logged every 5 seconds, in one csv file per day. This is for security only, as the microSD card is not easy to remove from the PLC casing. It should not be removed before the end of the experiment. | | |
| regulationSalinite()<br>*Only for Branched PLCs* | This function is responsible for the salinity regulation of the mesocosm. It sets the corresponding three-way valve position using a 0-10V analog signal. The function first checks if the regulation is in « manual override » mode. If so, it applies the override setpoint. If not, it reads the salinity measure in the mesocosm, compares it with the setpoint, and uses the PID settings to set the valve position. | | |

FACE-IT Mesocosm App

File   Settings   Maintenance   Data   About

**Ambient**

In Situ Data:
Temperature: 3.75 °C
Salinity: 34.85
Time: 2021-06-14 09:00:00

Ambient sea water
Pressure setpoint: 0.300 bars
Pressure measure: 0.302 bars
Valve: 70%

Cold sea water
Pressure setpoint: 0.300 bars
Pressure measure: 0.305 bars
Valve: 33%

Warm sea water
Pressure setpoint: 0.300 bars
Pressure measure: 0.311 bars
Valve: 60%

**Cntrl.**

**Trt. 1**

**Trt. 2**

**Trt. 3**

### Rep. 1

V3V: 45%
Flowrate: 7.06 L/min
O2: 114.01%
Salinity: 34.80
T°C: 3.75 °C

V3V: 0%
Fresh water Valve: 0%
Flowrate: 7.91 L/min
O2: 108.78%
Salinity: 33.21
T°C: 6.99 °C

V3V: 0%
Fresh water Valve: 0%
Flowrate: 8.31 L/min
O2: 106.06%
Salinity: 31.29
T°C: 9.03 °C

V3V: 100%
Fresh water Valve: 0%
Flowrate: 7.79 L/min
O2: 110.34%
Salinity: 35.06
T°C: 9.06 °C

### Rep. 2

V3V: 28%
Flowrate: 8.18 L/min
O2: 104.98%
Salinity: 34.90
T°C: 3.73 °C

V3V: 100%
Fresh water Valve: 0%
Flowrate: 7.36 L/min
O2: 106.04%
Salinity: 33.24
T°C: 7.12 °C

V3V: 89%
Fresh water Valve: 24%
Flowrate: 7.36 L/min
O2: 101.44%
Salinity: 31.50
T°C: 9.09 °C

V3V: 100%
Flowrate: 8.19 L/min
O2: 109.17%
Salinity: 35.16
T°C: 9.11 °C

### Rep. 3

V3V: 100%
Flowrate: 7.85 L/min
O2: 109.88%
Salinity: 34.71
T°C: 3.73 °C

V3V: 100%
Fresh water Valve: 22%
Flowrate: 8.38 L/min
O2: 114.73%
Salinity: 33.33
T°C: 7.13 °C

V3V: 100%
Flowrate: 7.65 L/min
O2: 101.49%
Salinity: 31.36
T°C: 9.05 °C

V3V: 0%
Fresh water Valve: 0%
Flowrate: 8.42 L/min
O2: 115.45%
Salinity: 35.15
T°C: 8.93 °C

Condition 0 set points
T°C:   3.75 °C

Condition 1 set points
Salinity: 33.29
T°C:   7.05 °C

Condition 2 set points
Salinity: 31.47
T°C:   9.05 °C

Condition 3 set points
T°C:   9.05 °C

STOP

Connection Status: Connected      Last updated: 6/14/2021 11:59:08 AM UTC      Experiment is running normally

FA�CE-IT
Arctic Biodiversity & Livelihoods

**Figure A3.** Application interface displaying real-time monitoring of ambient conditions as well
and control (Cntrl.), and treatment (Trt.) conditions for each replicate (Rep.) in each mesocosm.





















**A4. Menu bar of PC application**
From the interface, the user sets the temperature condition and associated salinity offset, IP
address and logging parameters, sensor calibration settings, and nominal pressure (Fig. A4).
Within the menu bar several tabs permit the setup of the project: file, settings, maintenance, and
data. Under 'file' the system can be manually connected to, or disconnected from, the PLCs.
Connection is usually maintained automatically. The 'settings' tab displays the application and
experimental setting options (Fig. A4.1 a – c). All the settings of the project are stored on the
computer (found in 'application settings') that is running the application, which include:
i.     *Master IP address*: The IP Address of the Master PLC (centralizing all the data).
ii.     *Data Query Interval*: Frequency of queries from the application to the master PLC.
iii.     *Data Log Interval:* Number of minutes between logs to file.
iv.     *Data Base File Path*: Directory and base filename of the csv data files.
v.     *FTP Username, Password, Path*: FTP settings for sending the data file every hour.
vi.     *InfluxDB Settings*: For Live Monitoring and local storage of the data.
Under 'experimental settings', the programmed specificities and regulation of the treatment
conditions can be adjusted. This includes programming the nominal pressure (all main inflow
lines), temperature and the salinity-temperature relational equation (on a different tab selected
from dropdown), as well as adjusting the $K_p$, $K_i$ & $K_d$ coefficients for the regulation (see section
2.3.1). The nominal temperature is provided by the data received from the ferry-box, however
this can be overridden if needed. The « Save to PLC » button sends the values to the
corresponding PLC and saves the data, while the « Load from PLC » button loads the settings
from the PLC. For the purposes of this experiment, the nominal salinity was calculated based on
a delta salinity for treatments 1 and 2 which were derived from the linear relationship with
temperature (see section 2.3.1). This can also be overridden if needed by selecting the manual
override box.

The 'maintenance' tab is where sensor calibration and communication 'Debug'

operations can be executed (Fig. A4 d, e). Calibration can be performed for each sensor deployed
in each mesocosm, and uses a 2-point calibration for temperature and % oxygen. The salinity
calibration is done by setting the conductivity value corresponding to a temperature of 25°C
rather than the *in situ* measured temperature. The conductivity value is programmed as $\mu S\ cm^{-1}$.
The communication process for sensor calibration is between 5 to 10 seconds. The final option in
the menu is the 'data' tab which displays the historical and live data. The historical data can be
interfaced to an html site if desired.

**a.**

AppSettingsWindow — □ ×

Master IP Address  172.16.253.10
Data Query Interval  1
Data Log Interval  5  minutes
Data Base Filepath  C:/Coco/CO2/FACEIT/NYA_Meso_Expe

**FTP**
Username  coconico2
Password  C0q2On0c@2021
FTP Directory path  ftp.obs-vlfr.fr/FACE-IT/Data_FACE-IT/

**InfluxDB**
Webpage  http://localhost:8086/orgs/de9bed37a0a3477/dashboards/077b31f0eb0d020000?lower=now%3A%2F%2F...
Token  eURNgzrqnoAEo6gOazgSW-QrPkwoeOSWGamvgn9puXTgUz1yG3Ez-OTnhU_JK6X3JamAzxrHZ5x+L6flgv=
Bucket  FACEIT
Org  CNRS

Save    Cancel

**b.**

Experiment Settings — □ ×

Control Condition  ‹ ›

**Pressure regulation**
Pressure setpoint  0.00

Kp  0.00
Ki  0.00
Kd  0.00

☐ Manual Override  0  %

**Temperature regulation**
Temperature setpoint  3.75

Kp  0.00
Ki  0.00
Kd  0.00

☐ Manual Override  0  %

Load from PLC    Save to PLC    Cancel

**c.**

Experiment Settings — □ ×

Condition 1  ‹ ›

**Salinity regulation**
delta Salinity =  -0.5458  x Ambient T°C +  0.4903    Update
delta Salinity setpoint  -1.56
Salinity setpoint  33.29

Kp  10.00
Ki  50.00
Kd  0.00

☐ Manual Override  0  %

**Temperature regulation**
delta T°C setpoint  3.30
Temperature setpoint  7.05

Kp  10.00
Ki  100.00
Kd  500.00

☐ Manual Override  0  %

Load from PLC    Save to PLC    Cancel

**d.**

Calibration — □ ×

C0  ‹ ›    M0  ‹ ›    Temperature  ‹ ›    Measurement:  3.72 °C

Calibration standard 1
Calibration standard 2

Factory reset

Set Offset    Ideally a value between 0°C and 5°C
Set Slope    Ideally a value between 20°C and 25°C

Please be patient. The calibration process can take a while. If the measure is not updated after 30 seconds,
you are allowed to re-click on the button. (Sometimes a double click does the trick)

**e.**

Communication Debug — —

**Request sent:**
{"command":1,"condID":0,"senderID":4}

**Response received:**
{"command":3,"condID":0,"senderID":0,"time":1623682025,"data":
[{"MID":0,"temp":3.711153,"cond":52891.537,"sal":34.81795,"flow":7.139935,"salSPID_pc":53,"tempSPID_pc":100,"oxy_pc":11
13.5461},
{"MID":1,"temp":3.717137,"cond":52892.36,"sal":34.87269,"flow":7.988964,"salSPID_pc":115,"tempSPID_pc":100,"oxy_pc":10
4.1252},
{"MID":2,"temp":3.737186,"cond":52676.52,"sal":34.71262,"flow":7.626548,"salSPID_pc":82,"tempSPID_pc":100,"oxy_pc":10
9.3133}],"regulS":[{"cons":10.3},"regulT":[{"cons":3.75}]}


**Figure A4.** Operation windows for the application and experimental settings (**a-c**). These
windows are found under the 'settings' tab. Operation windows for sensor calibration and
debugging (**d, e**). These are found under the 'maintenance' tab.





















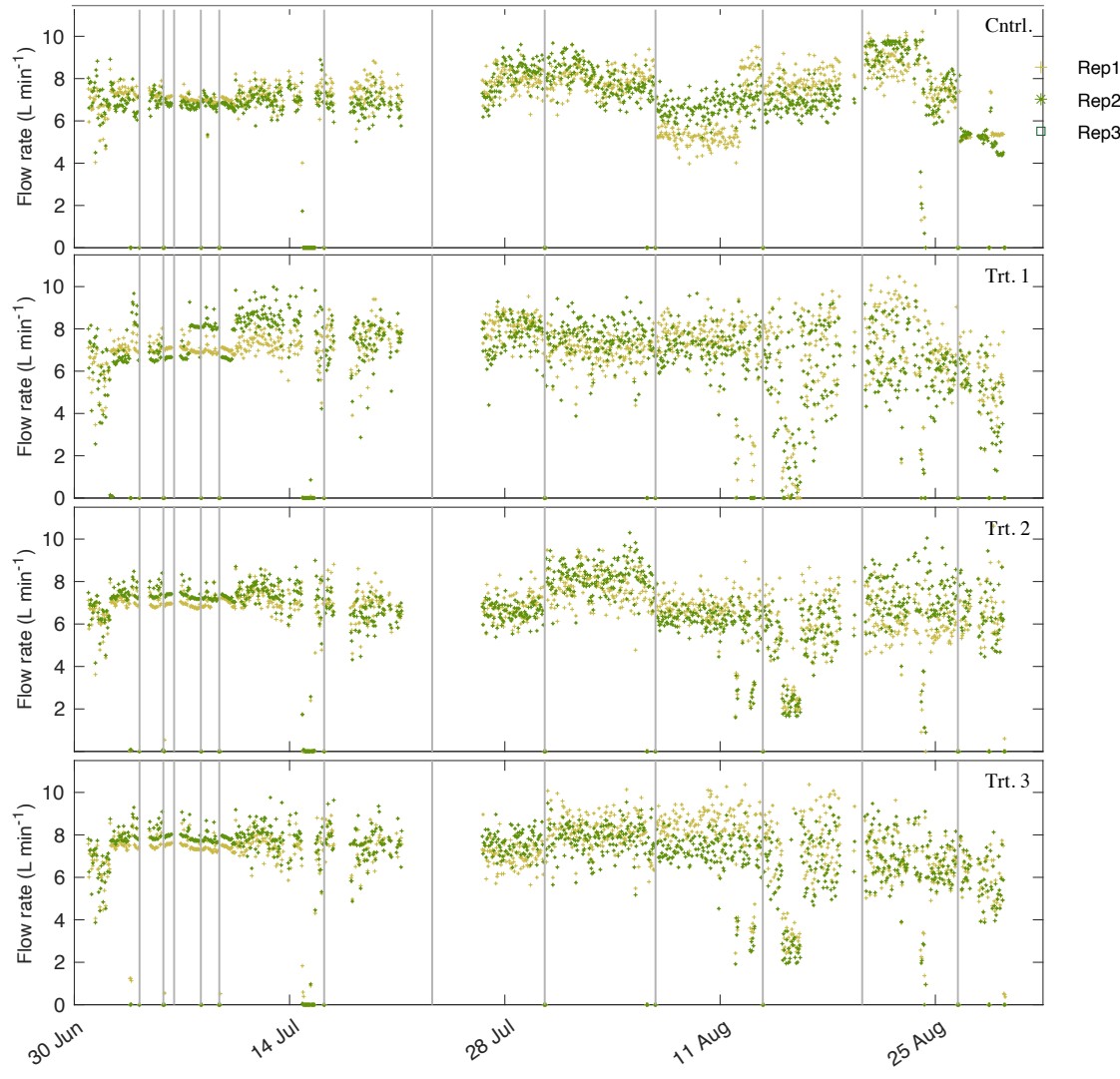


**Figure A5.** Flow rates for control and treatments 1-3 for the entirety of the system deployment.

Black vertical lines are when incubations were performed and the system shut-off for a period of

3 h. Flow rates went to zero at these times.





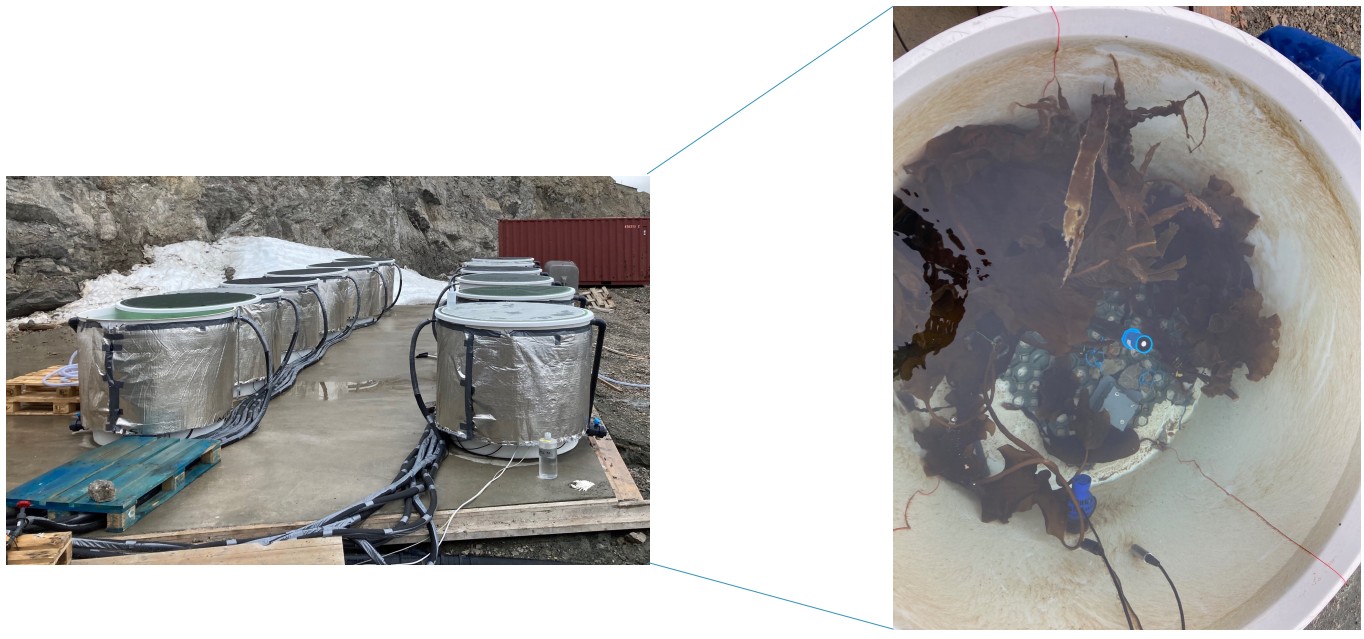


**Figure A6**. All 12 mesocosms are displayed (upper left photo) with the inside of one mesocosm
(right photo) showing the oxygen (silver) and temperature/conductivity sensors along with the
photosynthetically active radiation (PAR) logger (bottom right photo).






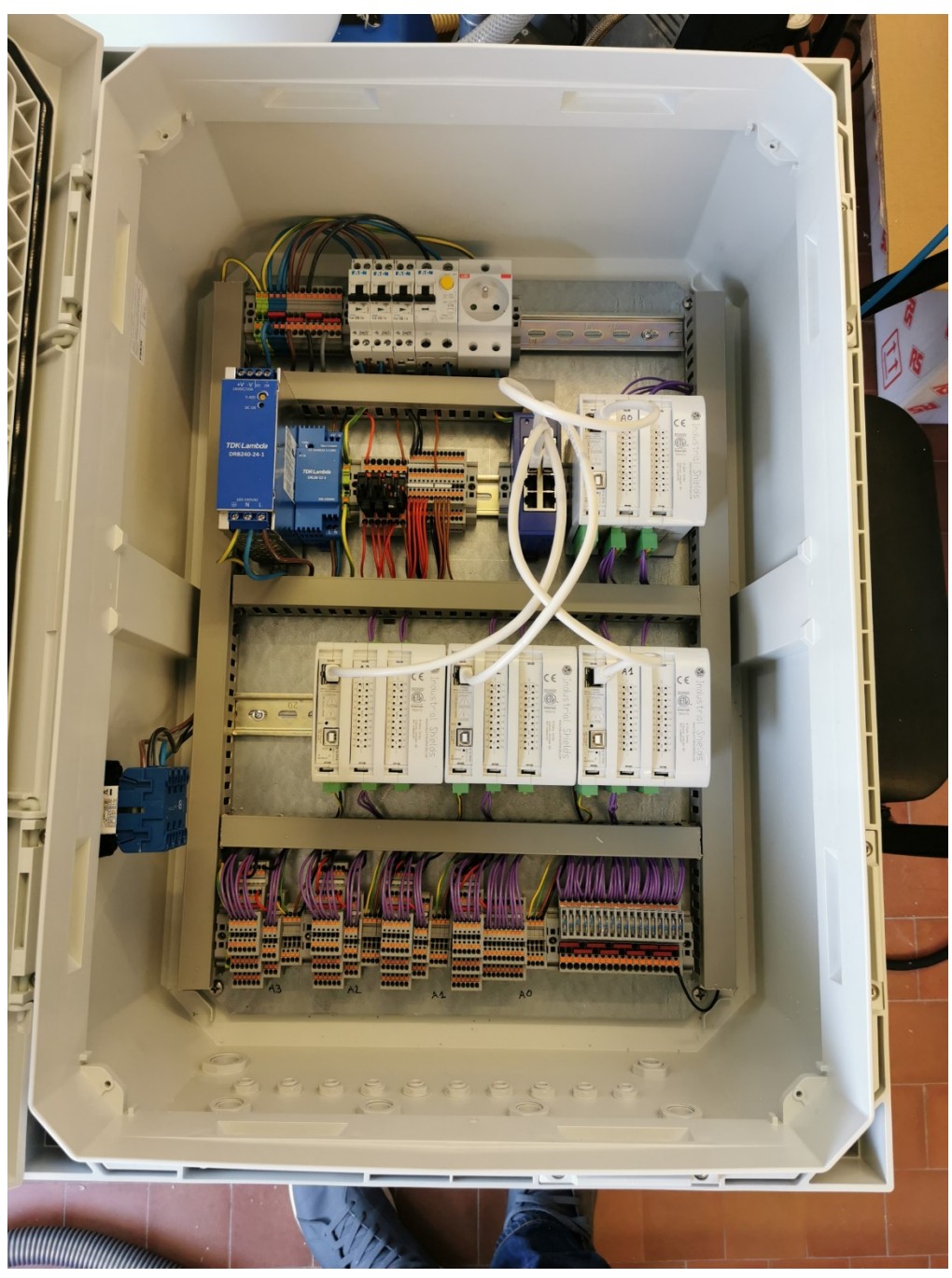


**Figure A7**. Electrical cabinet used for SalTExPreS




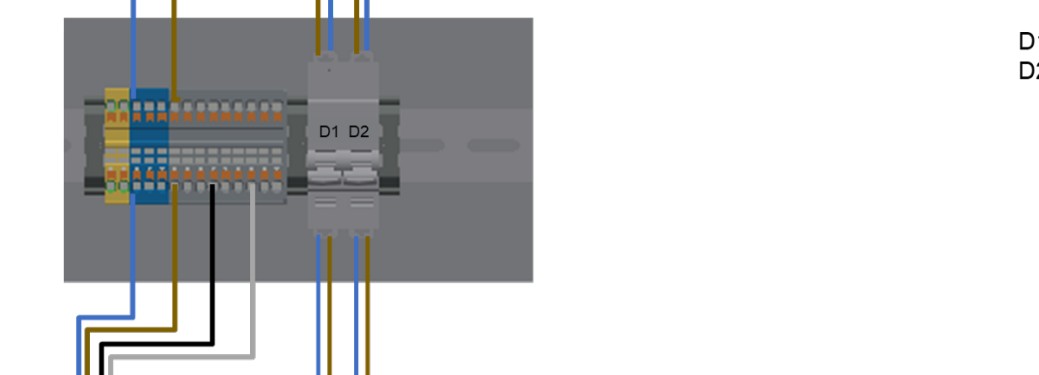

D1: Alim 24Vdc
D2: Alim 12Vdc


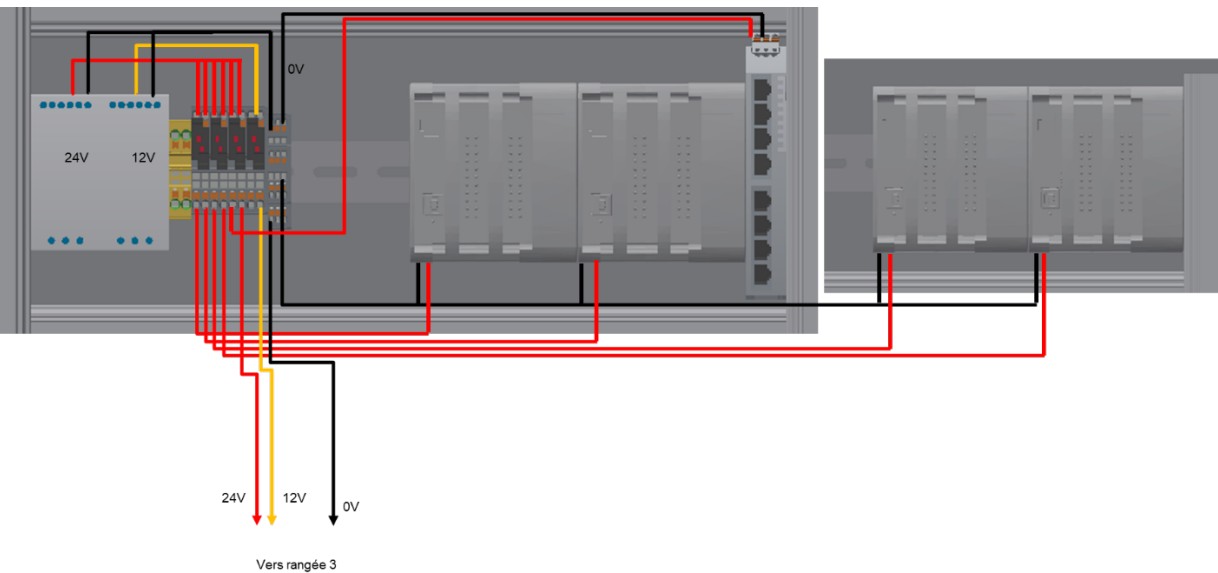


**Figure A8.** Electrical schematic for wiring within the electrical box.





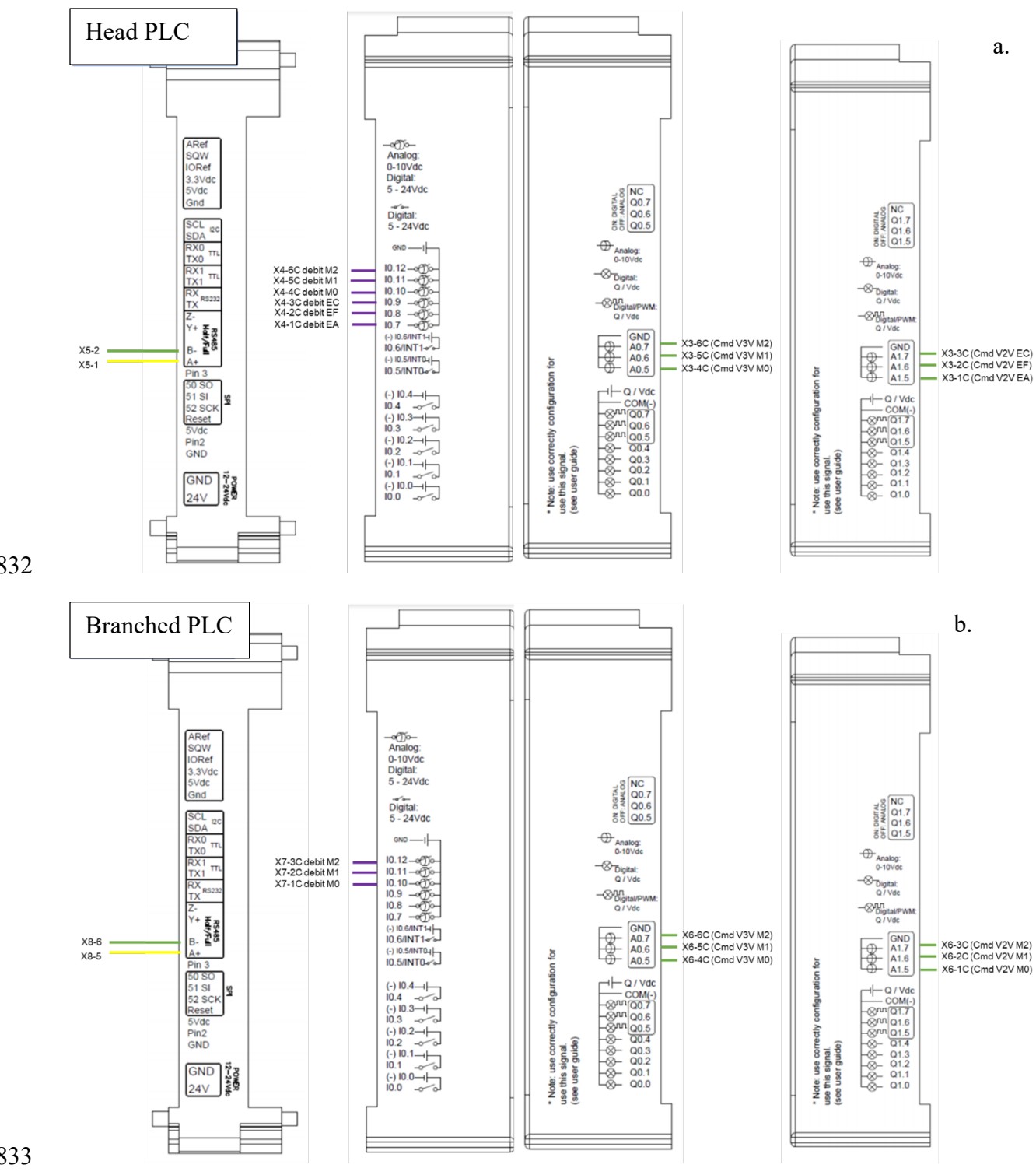



**Figure A9.** PLC controller diagram for Head (a) and Branched (b) operations.

