# Peer review of "Title: An Autonomous Flow Through Salinity and Temperature Perturbation Mesocosm System for Multi-stressor Experiments Author list: Miller, C.A.1,2\*, Urrutti, P.1, Gattuso, J.-P.1,3, Comeau, S.1, Lebrun, A.1, Alliouane1 5 6 S., Schle"

_EGUsphere, 2023_

## Author Comment (AC1)

Response to Reviewer 1:

**We would like to thank the reviewer for their time and comments. There are many relevant points of concern pointed out and we have tried to address these as best possible by adding clarity and detail. Some concerns by the reviewer we feel are out of scope for a technical paper, however, we have attempted to resolve these points to a degree we feel relevant. We hope that the substantial changes are sufficient as a revision to be considered for publication.**

**Please find below our response to all comments and concerns raised by the reviewer in bold typeset**.

Summary response and major concerns by the reviewer 1:

1) The introduction lacks a profound description of the Kongsfjord hydrology
   **We understand the concern about a lack of detail regarding the location of the setup. Since we focus more on the system which can be deployed in a variety of locations, we feel that a site description is best fit in the Methods section. Please see our added detail regarding site description in the revised manuscript, section 2.2.**

**2)** A referenced figure (S2) is missing: We apologize for not removing this residual figure reference as the original S2 figure was part of a previous manuscript iteration. It was removed due to poor image quality. **We have incorporated a new image of the setup in an updated figure 1.**

3) the three graphics in figures 1 and 2 mainly show the same but in detail are all different and inconsistent to an extent that it is almost impossible to really understand the setup. **We thank the reviewer for this comment and agree that the presentations in both these figures don't share coherence due to their schematical orientation. We have now combined figures 1 and 2 which we hope provides better detail about the experimental setup, as well as affording a visual of the system. Please see figure 1 in the revised manuscript.**

4) The discussion is mainly complaining about the problems that occurred in the beginning (and that they were solved) while it should show how strong the system is and what makes it special in comparison to other outdoor mesocosm systems (like e.g. Wahl et al. 2015). **We thank the reviewer for this comment and have rewritten the discussion to highlight the novelty and strengths of the system. We hope the newly written discussion projects this orientation**.

5) Finally, the structure of the paper as well as the language are not always clear and precise (you may ask a native speaker to review the text) and – as no biological data are shown – the proof is missing that the system can create a biological effect (I don't really doubt this but it has to be shown). **We have gone through the manuscript with finer detail to**

**assure lucidity throughout. We respectfully disagree with the reviewer that there needs to be evidence of a biological effect. As per suggestion by reviewer # 2, we have chosen to remove details about the biological component and focus more on the system itself. We agree with this suggestion as the aim of the manuscript is to show the functionality and assess the potential to manipulate environmental parameters, not to assess a biological response. We have, however, provided a figure below showing the continuous measurement of oxygen during an incubation to provide a visual of the biologically induced response. For further evidence of the experimental results that utilized this system we will provide reference to Miller et al. (community level effects) and Lebrun et al. (organismal level effects) as soon as the preprints are available. We will update this interactive discussion with the link to the preprint as soon as possible.**

[Figure]

**Incubation 1 for the control and treatment conditions.**

Response to line comments by reviewer 1:

LL 97-100 "The rapidly changing conditions in…": Please move this sentence up and add information to write a new paragraph where you roughly describe the abiotic oceanographic conditions in Svalbard fjords including the correlations between temperature, salinity and light conditions. **We have removed this sentence and included a version of the content in a broader description of Kongsfjorden in the manuscript. This new section (as suggested by the reviewer above) has been shifted to the beginning of the Materials and Methods section: 2.2.**

L 107: change to: '…treatments that represent expected future conditions…'. **We have reworded and changed this sentence. Please see lines 147 – 150 in the revised manuscript.**

L 109: change to: '…present at 7 m depth' (delete the 'a'). **This has been corrected.**

L109-110: change to: 'The treatments were realized by multi-stressor combinations of temperature, freshening and irradiance (Table 1).' **This sentence has been changed to reflect the reviewer's suggestion.**

L110-112 "The response of communities…": I understand that you want to publish your data in a separate paper. However, when you want to publish a technical note on your system, you need to show that your system works and that the treatment has an effect on the biology. Hence, I expect that you show at least some biological results of a pre-experiment or similar... Otherwise, I suggest to use the here presented data in the (extended) method section of the other manuscript - and not as a stand-alone paper. **We understand the concern of the reviewer to show the biological response of the organism(s) exposed to the treatments as a proof of application, however, we respectfully disagree that showing biological data is a standard procedure when describing an experimental system (e.g., Jokiel et al. 2014; Olariaga et al. 2014). We especially believe we can prove that an experimental system can work even if the simulated treatments do not have an effect on biology. The guidelines of a Technical Note also do not mention this within the scope of the journal. We feel that we have appropriately shown that the system can modify and regulate the salinity and temperature as described in the manuscript. To alleviate any concern about being able to stimulate a biological result, we point the reviewer to our response above which includes a plot of oxygen evolution during one of the incubations. We can also provide an update to this interactive discussion once the partner manuscript is published as a preprint.**

**Jokiel, P. L., Bahr, K. D., and Rodgers, K. S.: Low-cost, high-flow mesocosm system for simulating ocean acidification with CO2 gas, Limnology and Oceanography: Methods, 12, 313–322, https://doi.org/10.4319/lom.2014.12.313, 2014.**

**Olariaga, A., Guallart, E. F., Fuentes, V., López-Sanz, À., Canepa, A., Movilla, J., Bosch, M., Calvo, E., and Pelejero, C.: Polyp flats, a new system for experimenting with jellyfish polyps, with insights into the effects of ocean acidification, Limnology and Oceanography: Methods, 12, 212–222, https://doi.org/10.4319/lom.2014.12.212, 2014.**

L114-116: "The salinity offsets…": This comes here a bit out of nowhere. Needs explanation of the fjord conditions here or (better) in the introduction. **We have simplified this portion of the text and incorporated it into a more comprehensive description of the experimental design**

**from lines 156 to 163 in the revised manuscript. Section A1 now discusses the details of this calculation.**

L123: Fig. S2 is missing. If you don't have a good picture, please add a proper sketch to show the complete setup including the tanks (important to understand what you did). Also, the readers need to know how the treatments were distributed across the mesocosms... **We apologize for this discrepancy and have removed this figure reference. A photo of the system can now be viewed in the newly created figure 1 in the revised manuscript.**

Paragraph 124-131: Move this paragraph down after the whole system description (as a new paragraph 2.6). **We have removed this section completely as it was determined not to be very relevant to the system performance. This decision was made based on the newly written methods section and comments from reviewer 2.**

L124-125 "…experimental unit (i.e. a single mesocosm)…": What are these? Describe the mesocosms first, and then the organisms you put in. **This description has been removed in the revised manuscript.**

L139-131 "…with total biomass weights ~400 g for urchins and < 150 g for snails and seastars…": Please give the numbers of animals per tank! (very difficult to imagine the community from these weights...)! And: dry or wet weight? **This information has been removed in the revised manuscript.**

L137-138: You give the names and manufacturers of all the other devices in the table (which is actually table A2) also in the text. Why not these sensors, here? Please be consistent! **We have added the names and manufacturers to the in-text reference.**

L138: Its table A2! **We apologize for this discrepancy and have corrected this table reference.**

L146: Was the water temperature in the sub-header tanks also measured / logged? If yes, mention here! **The temperature was not measured in the sub-header tanks, it was only measured *in situ* at the COSYNA station and in each mesocosm. The text has been updated on lines 212 – 223 to clarify this point.**

L147 'constant flow': give flow rate. (or add: 'see below'). **We have reworded this sentence to describe the maximum flow rates rather than the flow rate to each mesocosm. We describe this later in section 2.3 of the revised manuscript.**

L154: What are 'weekly whole-system incubations'. Explain! **Given the comprehensive restructuring of the manuscript and removal of the biological description of the experiment we feel further explanation regarding this point is not relevant in this manuscript. This sentence has been revised to point the reader to these methods and results elsewhere. See lines 314 – 316 in the revised manuscript**.

L157: Can you please give some information on the quality (alkalinity) of the tap water on Swalbard? Is it ground water? Or from melted ice? Background: in some regions the tap water has very high alkalinity and significantly changes the water chemistry when used to reduce salinity... **The tap water in Ny-Ålesund is sourced from the Tvillingvann reservoir. Throughout this experiment we took several alkalinity measurements. For three consecutive weeks the average alkalinity in the control condition was: 2240, 2221, and 2202 and 2049, 2053, 2032 umol kg$^{-1}$ in the treatment 2 condition, which had a salinity approximately 5 units lower than the control (i.e., 34 to 29). The difference between these three consecutive weeks in alkalinity were 190, 168, and 169 umol kg$^{-1}$. Recently published data in Gattuso et al. (2023) reports a TA-salinity relationship of $A_T = 47.6 \times S + 643$ for a salinity range from 32 to 35 with an RMSE of 17 umol kg$^{-1}$ in Kongsfjorden. Using this equation, we estimate an alkalinity difference of 238 umol kg$^{-1}$. While this estimate is higher than our observed changes, our measured values were reasonable indicating no need for concern. Further, elevated alkalinity in freshwater is irrelevant for the functioning of the system described.**

**Gattuso, J.-P., Alliouane, S., and Fischer, P.: High-frequency, year-round time series of the carbonate chemistry in a high-Arctic fjord (Svalbard), Earth Syst. Sci. Data, 15, 2809–2825, https://doi.org/10.5194/essd-15-2809-2023, 2023.**

And: If the mixture with the tap water was done after the temperature adjustment: Why did this not change the temperature? **The tap water was warmer than the fjord water. The freshwater was mixed alongside the incoming temperature adjusted water. However, the system would regulate the inflow of warmed water to compensate for any temperature change induced by the freshwater supply. We have added a brief explanation of this in the revised manuscript. Please see line 200 – 202 in the revised manuscript.**

L158: add 'by' between 'rates' and 'using'. **This section has been rewritten.**

L166-167 "The inlet which is located…": Please describe better in detail or give a map. It should be possible to understand the position of the Ferrybox relative to the point where water is sucked-in for this experiment without the need to click a link... **We have added a detailed description of the flow-through system which is now easily trackable in the new figure 1 and section 2.3. We specifically report the position of the FerryBox (see lines 215 – 216 in the revised manuscript) and make clear the inlet reference is referring to the header tanks.**

L171-172 "Regulation was maintained via regulation flow valves utilizing…": change to 'Nominal values were maintained with regulation flow valves by utilizing …' **This part has been eliminated and a variation is now present in the newly written 2.3 section**.

L176: Explain 'PID'. Never use abbreviations without explaining them at 1st mention! **We apologize for this discrepancy and have added an explanation of the abbreviation as proportional integral derivative.**

L178 "…using a software PID controller": replace 'a' by 'the' **This has been changed from "a" to "the".**

L179-180 "…in PoE mode (proportion on error).": Explain what it does, e.g.: '…(proportion on error), which…' **We have corrected this to state 'Power over Ethernet" and have added a description as suggested by the reviewer. See lines 647 – 649 in the revised manuscript.**

L180: replace 'measure' by 'measured value' **This has been corrected.**

L181: replace 'setpoint' by 'nominal value' **This has been changed to 'nominal value'.**

L191 "…to the mesocosms.": Add: (Fig. 1 and 2) **We have added the figure reference.**

L193: This is confusing: the three sensors are the pressure sensors of the three lines (ambient, cold and warmed), right? And Kp, Ki, Kd are the coefficients which are described above and which are realized on the software level for each sensor (I guess). Clarify! **We have clarified this in the text to make clear that reference to the sensors are different than the coefficients described earlier. See lines 650 – 654 in the revised text.**

L196 'hand-crank valves': can these be seen in Fig. 1? Figure 1 is missing a proper figure legend describing all different valves. The hand-crank valves are missing in figure 2(?) **We made a new figure 1 where the hand-crank valves can be seen and are labeled accordingly.**

Figure1: In figure 1, the flow sensors (I guess) are placed AFTER the hand-crank valves. There are two pressure regulation valves for the ambient seawater...? The temperature regulation valves are labeled 'control' and 'treatment'. Harmonize! **We have made a new figure 1 to align with the flow of the system.**

Figure2: I believe you mixed-up the labels for ambient water and warmed water inlet. **This figure has been eliminated.**

Figures1 and 2: Overall, figure 1 and the two pictures in figure 2 don't fit very well together . EG in the photo one can see that the ambient seawater line in reality came from the other side. And not all valves are seen in all pictures and they are labeled differently and have even some errors. This makes it very difficult to understand your setup. The reader gets the feeling that you rather show the pictures because they are beautiful and not to help us to understand what you did. One even may think you didn't understand the figures yourself…

I recommend to maybe show only 2 of the 3 pictures and absolutely make sure they show the same (with same labels).

**We have made a new figure 1 which shows a more detailed diagram of the flow of the system.**

L 203: replace 'a single' by 'an individual' **This has been replaced.**

L203: 1 – 3, respectively. **This has been added.**

L206: branched (no capital B) **This has been corrected**.

Figure A2: tThe color of the digital communication is different in the legend (red) than in the figure (black). The lines are too thin to easily see the colors. **We have corrected this and made the lines bolder.**

L209: sending it to an FTP server. **The word "it" has been added.**

L210: replace 'made' by ' ensured' **This change has been made**.

L211: End sentence after 'cables': '…cables. The communication between the PLCs…' **We have made this correction.**

L212-214: …using a half duplex RS485 (2 wires) protocol, with an analog 4-20 mA and an analog 0-10 V signal, respectively.' **We have made this correction.**

L215 "…security switch": Add '(Fig. A7)' **We have made this change.**

L220-224: I'm not familiar with C++ and, hence, cannot comment on this section. **We have kept this description; however, this portion of the text has now been moved to the appendix.**

L234: replace 'valve-open percentage' by 'valve opening percentage' **This section has been rewritten and now states "valve opening position"**

L242-244 "The microSD…": Delete sentence**. This sentence has been rewritten. Please see lines 239 – 240 in the revised manuscript.**

L246: "2.5.1 Menue bar": Delete, as there is no 2.5.2: **This has been reassigned its proper section number.**

L247: add 'of the PC application' (or similar) after 'menue bar' **We have added this suggestion.**

L249-250 "There is also an exit button…": Commonplace: Delete. **This sentence has been removed.**

L266-267 "Loading should be done…": ??? Weird sentence that causes more irritation than that it helps. Delete (I guess)… **This sentence has been deleted. We agree with the reviewer that this does not aid the reader.**

L285-289 "The control condition..": Sentence doesn't work. Rephrase. **This has been rephrased.**

And: Actually, I would prefer to see the figure of the true temperatures  in the system (fig A5) first (and have it discussed) -  and then the figure with the deviations from control (fig. 3). **We**

recognize the reviewer's suggestion but disagree with where this figure should appear. We feel that the logical flow of the results section in the manuscript is appropriate as is: presentation and discussion of the regulation of the control condition followed by the performance and regulation of the treatment conditions in the following section. We have however, moved the figure out of the appendix and placed it into the main part of the manuscript and refer to this figure sooner in the text.

L191-305: Problems with the FerryBox and the 10m pump should Imo be placed into the discussion (separate section on technical problems (which you solved)). Also, as I see it, your experiment only started after you fixed the problem with the 10m pump. Instead of complaining about all the technical problems you had, you should show some biological data, here. **We agree with the reviewer regarding the placement of the technical issues and have moved them to the discussion while cutting extraneous detail. We disagree, however, that the experiment started after the 10 m pump came back online as conditions were actively being regulated before this time. Importantly, there was a stepwise increase for the treatment conditions during this period, which we feel is very much a part of the performance and experimental design.**

**Given the newly written methods section and the removal of the biological portion of the experiment, we do not see the benefit to include this in the manuscript as it will be published elsewhere. We refer the reviewer to our response above regarding this initial concern as well as to the above plot showing an example of an incubation ground truthing the biological response of the experiment.**

L310-311: 'the entirety of the planned experiment.': Repetition (in the bracket of the same sentence)! **The repetition has been removed**.

L316 "Due to technical…" : Same as above: move these technical issues with the temperature and salinity regulation to discussion section on technical problems! I mean: as can be seen in your figures, after you fixed the problem with the 10m pump, the system overall worked pretty well. But in your results, you  are mainly complaining about technical problems. This is not very nice to read and almost completely hides what you achieved. **We agree with the reviewer's comment and have rewritten portions of results as well as moved descriptions of most technical issues to the discussion.**

L323: '…(Table 2) were due to various anomalies: In the event…' **This sentence has been removed.**

L385: '…are examples of how improvements that were made during the experiment will allow for a more robust deployment in future.' **This sentence has been removed in the new version of the manuscript.**

L388-389: I suggest that you show some biological data from deployment 1 in this paper to show that your system had an biological effect - and use the 2nd deployment for your 2nd paper which then deals with your actual biological questions... **We feel that the suggestion by the reviewer regarding the biological data is not pertinent to the description of the system in this**

**manuscript. We have made many changes to this manuscript which focus on the deployment and operation of the system while significantly reducing any component of the biological experiment performed.**

L390: attempts **We have changed 'attempting' to 'attempts'.**

L390: 'mixing 'chilled' with 'ambient' seawater' **We have changed the placement of 'seawater' in this sentence as per the reviewer's suggestion.**

L425-428: Who programmed the software? **The second author Pierre Urrutti programmed the software. We have added this to the contributions section**

Figure 3: 'Regulation of the temperature offset from the set control value during…' **We have rewritten this caption for clarity: 'Regulation of the mean temperature offset for all conditions, including the control offset from the FerryBox, and the three treatment offset values from the control condition.'**

Figure A1: Replace scenario numbers by treatment numbers/names and add scenarios in only in the figure caption. **This figure has been remade with a new corresponding caption. Please see Figure A1 in the revised manuscript and the associated caption.**

Table A2: I understand that you don't want to mention the city and country of the manufacturer in the table as this might take a lot of space. However, I think this information is important as companies sometime change names are bought by other companies etc. - and this can cause confusion. Therefore, please give the city and country of each company in the table, in the table caption or in a separate table...**We agree with the reviewer on the importance of providing detailed information on the origin of manufactured products used in the system. The city and country of the manufacturers has been added to the table.**

---

## Author Comment (AC2)

Response to reviewer 2

**We would like to thank the reviewer for the very thorough comments and suggestions on how to improve the content for the reader's benefit. We agree with the reviewer in nearly all instances in this regard and have rewritten the manuscript to follow the format suggested by the reviewer. We hope that these changes are acceptable for publication.**

**Please find below our response to all comments and concerns raised by the reviewer in bold typeset.**

Summary response and major concerns by the reviewer 2:

It takes a lot of work from the reader to understand how the system works. The fundamentals are buried in a lot of extraneous detail, and references to many figures that overlap in their content but are not unified in how they are presented and explained. It needs to be rebuilt starting with the main ideas that in interested reader would need to know including to following topics which map roughly into the introduction, methods, results and discussion.
-Under what circumstances is the system is most useful? (without implying that other less complex systems are no good)

1) What are the fundamentals of how the system works? (without getting into the weed too much). **We have added an "operational concept" section (2.1) which describes the capabilities of the system in a simplified manner.**

2) How does the system perform? (with more emphasis on the second use in the heat wave experiment when the improvements were in place and less on the difficulties of getting to that point.). **We have focused on the second deployment description in the newly written discussion to focus on the benefits of the improvements made.**

3) What are the best applications of the system and what are the main considerations for someone who might want to use to be able to make the best use of it? (without a tale of woe about things that went wrong in this build apart from those points). **This is summarized in the newly written 'operational concept' section (2.1) which gives a basic description of uses and capabilities. We also highlight the system potential in the newly written introduction.**

Specific Comments by reviewer 2:

The introduction would be stronger if it included a more robust (and generous) description of previous multi-stressor systems, particularly for salinity and temperature. There is no need to say these systems only had static changes or course automation. Instead, describe a couple of them without the devaluing adjectives. You can say systems include static conditions which can be appropriate for low variability systems, and that automation in existing systems is based on set patterns of fluctuation, on scales of hours to days (or something), producing regular pre-determined variability. This kind of system could certainly replicate tidal variability and manageable human intervention could simulate spring and neep tides. There is nothing intrinsically inferior in these approaches. You need to make the case for why a system that creates and offset from naturally observed variability is valuable. (I believe it is, but you need to make a stronger case for it.)

The introduction should also include some information about the magnitude of the variability in salinity and temperature and the time scale of these changes so the reader can better understand why an autonomous system is needed instead of manual control.

**We thank the reviewer for their comment and agree that the focus of the introduction be reoriented. We have substantially changed the introduction to reflect the comments by the reviewer including reference to other experimental systems, removing critiqued language of other systems, and a more detailed description of the variability observable in nearshore environments.**

It's awkward for the methods to begin with the kelp experiment since that is not covered in this manuscript. Just talk about the system here, not the experiment if the experimental data are not also presented. Or at least don't lead with the experiment.
**We have cut down on the amount of detail given to the biological experiment and focused more on the system itself. The methods section now begins with details about Kongsfjorden hydrography and explicit details about the biological experiment have been removed since we do not present these data.**

Your approach for getting the salinity offset seems unnecessarily complicated. First of all, is there some reason to believe that the salinity/T relationship is non-linear? The summer data don't clearly indicate this, and could be fit to a linear equation (though with all the projection points on the graph it's kind of hard to tell – projected data should not be shown as points here.) But if we accept it's non-linear, which it certainly could be, then why fit new linear equations for each treatment when these are simply replotting the results of the projection from your quadratic equation against an offet temperature? (Or at least I think this is what you have done – though I can't be sure since it's not really explained.) This all needs to be better explained and better justified. Is the software only capable of using a linear model to generate the target salinity? **We agree with the reviewer that the original description was convoluted and unclear. We have reduced the text in the manuscript to point out that the salinity offsets were derived based on *in situ* temperature correlations with salinity in the summer of 2020 at 11 m in Kongsfjorden. We have provided a detailed description of the calculations now in Appendix A1 and have changed figure A1 as well. We feel the newly added description provides clarity on how these calculations were made as well as a strong association with what is now presented in Fig. A1.**

The explanation of the system would benefit from an overview at the start. For instance, you show pressure regulating valves in figure 1, but when the reader first looks at this figure when it is called out, their purpose is unclear and you have to wait a while till the pressure regulation section to understand what they are. And even then, while there is a lot of detail, the basic function of the valves is not that clearly explained.  Just start with an overview in the experimental system section that very briefly goes over the basic function of the whole system.  Anything that is important enough to be in the schematic figure 1 gets a brief explanation here.  And one schematic figure is enough, or a photo but not both and the photo is nice looking but harder to label and explain. **We thank the reviewer for this suggestion and have added a new section to the methods that gives a brief overview of the system concept. See the new 2.1 section in the revised manuscript.**

There are too many references to the appendix material in the main text. This is an issue when the text is not that easy to understand without reference to these, and it's inconvenient to make these references into an appendix.  You should consider what the general reader who is interested in the method in a general sense need to know.  Keep that in the main text.  Things that someone who might be interested in building a similar system of their own can refer to more detailed information in the appendix.  This will mean moving some parts of the text to the appendix.  For instance – the whole menu bar section is more like a user manual for the software.  This is just not needed to understand the system and its merits.  It should be moved to supplementary materials. **We thank the reviewer for this comment and have streamlined what is necessary to keep in the main text and what should be moved to the appendix. We have now moved many parts of the methods (Temperature and Salinity Regulation, Pressure and Flow Regulation, Automation, Software development) to the appendix. The methods section now retains the necessary information to understand the system without extraneous details.**

The figures need to be consolidated to just what is needed to convey the material.  Too many figures make it difficult to navigate and labeling varies in the text and between figures.  All this needs to be streamlined. **We have combined figures 1 and 2 to aid the reader in understanding the system. The appendix maintains the previously presented figures, but these figures are now referenced in the appendix so as to not complicate the main text.**

Another example of a section with too much detail is section 3.2 in which you dissect the reasons for different instances of the system not meeting the set point.  This needs to be greatly simplified and framed not as an explanation for deviations in the data but perhaps a very brief record of improvements that were made to the system and an account of areas that the user must pay attention to for optimal operation such as clogging or loss of pressure in the system.  The point of the paper should be about how the system can best be used, not looking under the microscope at every bump in this deployment of it. **We agree with the reviewer's comment and have reduced significantly the description of technical issues and have moved them to a brief part in the discussion. Section 3.2 has now been streamlined to highlight the performance of the system as suggested.**

Figure 6 appears as a kind of add on, but should be featured more prominently.  Why spend the whole thing focusing on the shake-down use of the system in which the bugs were worked out, and then not really talk about the one where it seems to have worked really well?  Yes, all the work happened in the first experiment but the data from this heat wave application are more compelling as to how this system can be really useful and powerful. **We appreciate the insight and comment by the reviewer. We have chosen to retain the presentation of Figure 6 in the discussion as it is well suited to show how the adjustments made to the system improved its**

**functionality. Since we moved the text describing the technical issues into the discussion, which were followed by suggestions on optimal operation (as per the reviewer's previous comment), the presentation of Figure 6 at this junction seems appropriate. Please see lines 326 – 331 in the revised manuscript on how this figure was incorporated into the new version of the discussion.**

It is not explained how adding freshwater after the temperature is adjusted does not influence the ability of the system to meet temperature set points. **The tap water was warmer than the fjord water. The freshwater was mixed alongside the incoming temperature adjusted water. However, the system would regulate the inflow of warmed water to compensate for any temperature change induced by the freshwater supply. We have added a brief explanation of this in the revised manuscript. Please see line 201 – 202 in the revised manuscript.**

Wording in general is often not concise and phrasing is awkward especially in the use of passive voice, such as "Challenges… were able to be resolved henceforth". Just say challenges were resolved. **We have done a comprehensive rewrite of the manuscript and feel many of these adjustments have been made.**

Response to line comments by reviewer 2:

Wording is unclear.  Flow rates of chilled or heated ambient seawater and freshwater?  Not clear what is chilled and/or heated. **We have changed this sentence to make clear the intended message. See line 47 – 50 in the revised manuscript.**

Eliminate the start of this sentence about versatility.  It's unclear and redundant with the second half of the sentence. **We have rewritten this sentence. See line 59 – 60 in the revised manuscript.**

Why not just say extreme variation in temperature and salinity?  "physiochemical conditions" is vague.  Another statement can establish that these changes are physiologically and ecologically relevant if you like: **We thank the reviewer for their suggestion and considered changing this phrasing, however, these processes induce several changes beyond temperature and salinity such that we think physicochemical is appropriate.**

Awkward – need to add "how" species richness… or reword: **We have removed 's' from 'functions' to correct this sentence.**

This sentence is long and tangled.  Generally the use of the word "various" is a red flag. **This sentence has been split in two and reworded. See lines 84 – 89 in the revised manuscript.**

I don't understand what this sentence is trying to say, and if I did – it would only be one advantage. **This sentence has been revised. Please see line 99 – 103 in the revised manuscript.**

Not sure about this sentence either – not requiring constant human observation seems good and a clearly part of an autonomous system. That is what makes it autonomous. Doesn't seem like it needs a lot of explanation, and the caveats about new programs and rapid adjustments are muddying the waters here. **This sentence has been changed. See lines 103 – 105 in the revised manuscript.**

That acronym is a stretch. Do you really need it? **This acronym could be changed but we feel the current one doesn't really retract from the content of the manuscript**.

the word "following" is confusing here. Wordy overall. Just say Temperature anomalies from the X and Y scenarios were selected for this test of the system. **We have reworded this sentence to simplify the message. We have chosen to retain the brief explanation that the chosen conditions were informed by the projected temperature anomalies predicted for the Arctic as this adds relevance for the applied anomalies. See lines 158 – 163 in the revised manuscript.**

Figure S2? **This has been removed**.

(3 treatments, 1 control, x3 replicates) Just give the mean diameter to simplify. **We have provided the mean diameter as a suggested simplification.**

it is more standard to give the make and model of this kind of instrument in the text though a parts list as supplement is nice too. **We have added the instrument description to the text as well.**

This sentence is confusing and since the offet from in-situ conditions is key to the idea here, it should be highlighted and clarified. **This sentence has been removed and the description of the applied temperature offset to a measured *in situ* condition has been highlighted in section 2.2 as well as at the end of the introduction**.

Confusing- if the flow is constant from each header to the mesocosm then there would be no variation. **This sentence has been changed to express constant pressure and maximum flow rates for each sub-header tank.**

This makes it sound like you are guessing how much heat is added in transport. Make it clear that the chilled water is added by the control system to match measured in-situ conditions. **We have adjusted this sentence for clarity given the confusing wording suggested by the reviewer.**

What does this mean? Eliminate minutely, but even then its not clear. The inflow pressure of each of the water sources pre-mixing? It's not clear exactly what the pressure and the outflow rates tell you. **Briefly, the monitoring of pressure for each sub-header tank ensured that**

**sufficient flow rates could be maintained in order to meet setpoint targets. Outflow rates were measured to monitor the turnover in each tank. These were set to be 7 – 8 L min$^{-1}$. We feel this is now properly explained in the new 2.3 section.**

164- wordy and tangled sentence. I am not going to stop commenting on style and grammar at this point since there is going to be some reworking of the whole needed and more attention needs to go into the writing before that level of review is appropriate but this is a good example – it should read " Temperature and Salinity in mesocosms were tuned to match hourly in-situ readings from…" or similar. **This sentence has been removed and the section rewritten**.

How can the error be the integral and the derivative of itself? This just doesn't make any sense. I think you mean the error, it's derivative are factors in a prefit equation to determine the appropriate adjustment to the valve? **We apologize for this typo. We have made the correction in the manuscript to state: "The PID controller measures the difference between the measured value and the setpoint (i.e., the error). This calculates the position and adjustment of the valve opening by multiplying the error, the integral of this error, and the derivative of the error over time, ….|"**

Kp, Ki and Kd are coefficients (line 183) but here you refer to them as sensors, or so it seems. But also it seems like this is meant to be a pressure set point that is the same for the system regulating the 3 temperatures? **The latter comment by the reviewer is correct. This was meant to state the pressure regulation followed the same method as the temperature regulation. This has been changed in section A2.1 of the revised manuscript.**

Why would the flow to the mesocosm be adjusted? What is the goal of these adjustments? Does flow vary that much depending on the mixing valves that this is necessary? This section needs some explanation. **This section has been completely rewritten. The purpose of regulating flow to each mesocosm with a final hand crank valve is to regulate the turnover time in each tank. This should be clearer in the new 2.3 section.**

and 309 Eliminating part of the data because the system was not capable of correcting the issue is not really valid. Instead you should describe the start period, the ramp up of T but not include these in the time you cite the system was controlling conditions. Instead, report the period after the 10 m pump came back on as the time when the system was fully operational creating the experimental treatment conditions. **We thank the reviewer for their thoughtful comment, but we do not agree that part of the data was erroneously removed, nor do we necessarily agree with removal of regulation during the ramp up period as suggested by the reviewer. Data here were not omitted and we refer readers to figures 3 and 4 as well as table 2. We believe this error in the understanding of the text comes from the incorrect wording referencing table 2. We have only chosen to omit the calculation of the percent time of deviation greater than the mean (figure 5) during use of the 90 m pump due to an exaggerated deviation between the control temperature and setpoint during this period. Further, the temperature ramp-up period is certainly part of the design and, thus, necessary for reporting the performance of the system. We do not feel a need to separate this into three different conjunctions but rather report the results during the period in which the system was operating. We have, however, changed the text to correct the**

**potential misunderstanding readers may have of us removing data when utilizing the 90 m pump.**

and 320 Why report the general performance for Temperature first, then again together with salinity? Steamline this and either do T and then S separately or present together once. **The first section of the results refers specifically to regulation of the control condition while the second section refers to the treatment conditions. We can see where this may be confusing based on the topic sentence for both sections. We have revised the topic sentence for the second section to make this clearer. Please see section 3.2 in the revised manuscript.**

I said I wasn't going to comment on this kind of thing, but this sentence is another example of the kind of thing you need to eliminate from your writing. The sentence starting on this line has no information in it and also a grammatical error. **This sentence has been removed.**

Table 1/Figure A1 – If the salinity offets are negative, make that clear here. Also the equations to determine the salinity setpoints should be in terms of S, not y. In the figure, it is not entirely clear what you did. The colors are hard to distinguish but it looks like you fit a curve to summer data with T between 2 and 6 degrees and then predicted salinity values for the projected future temperatures using that curve. If you don't plot the points for the projections, perhaps this would be more clear. Just showing the range of T for each projection would be enough, or not breaking them out at all in this plot (A1a). Part b here also requires a bit of leap to understand what you did. This seems unnecessary. See comments in main text. **We have made mention of the negative values in the Table 1 caption as well as changed "y' to ' delta S'. For figure A1a, we have remade the figure to show the measured *in situ* relationship for which the 2nd order polynomial was fit. We have also added a more descriptive text to the figure (section A1) which better describes the process that was applied to calculate the salinity offset values.**

Figure 1- why is the ambient seawater not just in line with the chilled and warmed ones? The lines are crossing a lot for no reason in this figure. The sensors are really not adding anything meaningful here – only keep if you are going to include the feedback to the control system/valves conceptually in the figure but that might be hard without making it too messy. Figures 1 and 2 are really not needed. They seem to be giving two views of the same info. Make one that is more clear. **We have combined figures 1 and 2 into a more logical description of the system flow. Please see Figure 1 and caption in the revised manuscript.**

Figure 2- The role of T and pressure regulation valves is not clear. I am guessing the pressure regulation valves are back pressure regulators, to ensure that the pressure on the upstream side of the main control valves remains constant. You should explain that if so. Then the T regulation valves are using a pressure drop to measure the flow allowed through, which is determined by the control system and the feedback from the sensors. Explain it! **The pressure regulation has been better described in the new methods section (2.1 and 2.3). Briefly, the pressure regulators were put in-line to monitor the flow from the header tanks which first passed through a valve regulator. Pressure at this point needed to be maintained at approximately 0.3 bars to ensure sustained flow rates of $7 - 8$ L min$^{-1}$ to each mesocosm.**

Figures  in Appendix in general – there are so many of them.  This is an issue when the text is not that easy to understand without reference to these, and it's inconvenient to make these references into an appendix. **We have substantially reduced the references to the appendix and have moved a significant part of the descriptive methods to the appendix section as per your previous comment. We feel this has greatly improved the readability of the manuscript.**

Figure 3 – the flow rate heat map feature on this figure is not intelligible.  There does not appear to be any difference in color in the different values for flow rate.  I guess the shaded areas are when flow is greater than 2 l/m but the legend does not indicate that in any way, and the text and caption don't explain why this is and important threshold or how flow rate would influence the performance of the system. **The reviewer is correct, there is no color difference in the heat map. However, there was no intention to indicate these differences, but rather, to show when flow rates were $\leq 2$ L min$^{-1}$. We have added a more descriptive caption to indicate that the shaded regions are when flow rates were $\leq 2$ L min$^{-1}$. The 2 L min$^{-1}$ threshold indicates the minimal flow rate needed to avoid large deviations (> 2.0 salinity or °C). This threshold was determined by observation of the timeseries. This information also appears in the discussion section in the revised manuscript.**

Table 2 – The manipulated water section of this table takes a lot of space and doesn't really add much.  The info that convey is methods of how the treatments. **We feel that the "manipulated water" portion of the table assists the reader in determining why the table only shows salinity deviations for treatments 1 and 2. This is beneficial for the reader if they are quickly looking through the figures and tables while skimming the text.**

---

## Referee Report (RR1)

**2nd revision:**

**Comment on: "Technical Note: An Autonomous Flow through Salinity and Temperature Perturbation Mesocosm System for Multi-stressor Experiments" – Version 2**

Dear authors,

Dear editors,

Overall, the paper indeed improved significantly between versions 1 and 2. However, it is imo still pretty far from being publishable: The text is in parts still clumsy/sloppily and not very accurately written. This might be due to the 1st author being rather unexperienced, but at least two very experienced co-authors should also take care of a scientifically proper text.

Furthermore, the very important figure 1 is somewhat better than the previous figure - but still (due to poor execution), confusing and far from a professional sketch that the paper requires (and deserves).

Finally, I don't know if this is due to being unexperienced or active ignorance or not taking the reviewers serious: (At least) in three cases in the answer to reviewers comments the authors write that they changed the text according to the reviewers suggestion - but actually did not do so:

1) they still write 'setpoint' instead of 'nominal value'

2) they still write 'valve-open percentage' instead of 'valve opening percentage'

[Both are no major points, still they should not write that they changed it - and then don't...!]

3) In their responses to the 1st reviews, they advertise a figure on oxygen data from the mesocosms, which would have improved the paper, significantly. However, its missing in this version....

I don't know what to say...

Details (again, mixed minor and major):

L 72: '…will be affected when / while assemblages adjust…'

L 74:  'Methodological approaches to assesS and characterizE the responseS of organisms…'

LL 74-79: Sentence too long (and weird): Make a full-stop after the parentheses in L 77!

L 109: write 'above' instead of 'previously'

L 112: 'In THE initial deployment…'

L 143: replace 'shoal' by 'reduce'

L 144: '…in the Kongsfjorden environment…'

L 144–146: Make it two sentences: '… deployment of the SalTExPreS. It was placed on a concrete platform…'

L 148: Replace 'represent' by 'simulate'

L 149: Replace 'examined' by 'supervised'

LL 149-150: Rephrase. It sounds like the communities can be found at 7m depth for 54 days, while you want to say that the experiment took 54 days…

L 152: '- the details': delete 'the'

L 150–153: Move this sentence to the end of the paragraph!

L 157: '11m' confusing: You want to simulate a -7m community but measure the fjord conditions at 11m???

L 166: 7m or 11m???

L 171: Now its 10m???  Clarify!

L 172: '… that was tapped into an underwater intake pipe AND that fed a header tank…'

L 176: 'plumbed'…? What do you ant to say?

L 178 and L181: Dot missing after 'Fig'

L 182: '…from the tap which IS fed by the…' (guess this is still the case.)

L 185: delete last part of the sentence (',  where flow rates of  7 – 8 L…' ) as this is redundant to the previous information given.

L 186: replace 'incubations' by 'interruptions'

L 188 write: '(3 treatments and 1 control, each with 3 replicates)'

LL 191-192 write:  'Fiberglass insulation at the outside of each mesocosm reduced unintended changes in treatment water temperature.'

L 193: delete comma after 'warmed'

L 196: no hyphen between flow and line

LL 197-198: move '12 in total' to the end of the sentence : (12 in total, Fig. 1)

LL 203–205: redundant and commonplace: rephrase!

L 205: delete 'all'

L 206: delete comma after 'valves'

LL 206–207 just write: '…logged every minute and displayed on the user interface (Fig. A3).'

L 209 (and throughout the whole manuscript!): replace 'setpoint' by 'nominal value'

L 231: 1st mention of 'PLC': define here as 'Programmable Logic Controller (PLC)'!

L 231: '…informing ON proper communication…'

L 233 (and elsewhere): replace 'valve-open percentage' by 'valve opening percentage'

L 237: % is no unit of concentration! Write: '.., O2 saturation (%),..'

L 247, 249 and elsewhere: replace 'control condition' by 'control treatment' or just 'control'

L 251: comma after 'period'

L 251 'across the 3 replicates', clarify: does this mean this is the average off the deviation over all 3 control replicates - or was the average in each replicate less than 0.3°C?

L 251-254: replace 'was based on' by 'was achieved by'. Furthermore: This sentence doesn't make sense at all: The ability to read data does not help to keep regulation quality high (low deviation from the nominal value). The fast response of the system to changes in the ferrybox data is the key player here...

L 253-254: put last part of the sentence (after the comma) into parentheses.

L 262 write: '…situated at 90 m depth in the fjord was used from…'

L 263-264: write: '…at 10 m depth was repaired.'   (motor malfunction info is irrelevant)

L 275: replace 'deployment' by e.g. 'experiment'

L 281-282: delete: '…after the final incremental increase was programmed.'

L 2282-284, Rephrase: I don't understand this sentence…

L 303-304:  Please discuss: The ability to add warmed seawater also to the control system would allow for keeping the control stable even when using the 90m backup pump (in a future setup).

L 311: replace 'inaugural' by 'first'

L 317: replace 'permits the' by 'allows for the'

L 325, write: 'Since the initial experiment, we implemented a number of changes to improve the performance of the system which have been realized during a second experiment in …'

L 332-334, write: '…kept deviations in the 9 different mesocosms at < 0.5°C for 94% of the time (79% in the first experiment).

L 334: 'During the first experiment…'

L 335: 'largeR deviations'

L 336: '…flow rates OF < 2 L min-1…'

L 337: Simple software modifications

L 340-341, just write: '…data were maintained solved most of the issues.'

L 343, write: '…and clogging of the seawater inlet are issues that need to be addressed whenever…'

L 345: How do you prevent clogging or remove the stucked material? Can the pumping direction be reversed?

L 346-347, write …' independently regulate experimental conditions in a …'

L 350: replace 'its' by 'the system's'

L 351: new sentence after '(e.g. tidally).'

L 351: replace 'mimic a future scenario' by 'mimic future scenarios'

Figure 1:

- Sensor labels too small
- 'Control' doesn't make sense as the violet/dark blue line also supplies the 'treatments'
- missleading: are the yellow marked valves 'pressure regulation valves' or 'pressure sensors' (as stated in the text below)? Or aren't the yellow marked things in the photos the same as in the sketch?
- Black line: where does the freshwater enter the system (and where starts the arrow that points at the hose in the photo)
- Overall: using some colorful lines as the actual pipes/tubes of the system - and other colorful lines as arrows that point at the fotos is very confusing and generally unacceptable for a scientific sketch.
  I suggest that you use a number code (like in the foto - and write the same numbers next to the respective line or valve in the sketch.
- It is very important to be exact into each single detail in this figure!
- L 539: that lead to all 12 3-way regulator valves
- L 544: photosynthetically without capital 'P'

Figure 2:

- L 560-561: Not 'regulation': Never write what can be interpreted from a figure in the caption, but just what can be seen!  Just write: 'Mean temperature offsets of all applied conditions. Blue: offset of the control from the FerryBox. Dark green, light green and yellow: offsets of the treatments 1, 2 and 3 from the control, respectively.'
- L 562: end sentence after 'standard deviation'. (Short sentences are good!)

Figure 3: 'mean' values: averaged across how many single datapoints? resp. across how much time???

---

## Author Response (AR2)

**Dear Editor,**

**Please find below our point-by-point reply to the referee's comments. We are sorry that some issues remained in the previous version of the manuscript and believe that all of them have been addressed satisfactorily in the present version. The review process has improved the quality of our manuscript. We trust that it now meets the standard of publication of Biogeosciences.**

**Regards,**
**Cale Miller on behalf of all coauthors**

Overall, the paper indeed improved significantly between versions 1 and 2. However, it is imo still pretty far from being publishable: The text is in parts still clumsy/sloppily and not very accurately written. This might be due to the 1st author being rather unexperienced, but at least two very experienced co-authors should also take care of a scientifically proper text.

**We thank the reviewer for taking their time and making the effort to provide significant feedback aimed at improving the paper. We feel the manuscript has improved significantly from the overall review process. We carefully considered and respected the comments from all reviewers. The significant rewrite in the first iteration should reflect our dedication to this. Subsequently, we have enhanced the manuscript further, as detailed below in the following changes.**

**However, we believe it is important to state that about 75% of the comments from the reviewer pertain to changes in grammatical structure for sentences that were not grammatically incorrect. In this regard, we truly appreciate the time and effort devoted to enhancing the manuscript, but we feel that as long as the grammar is correct, it is at the discretion of the authors to choose which writing style is most fitting, while considering the reviewer's comments to improve clarity, which we have done. Thanks to the thoroughness of the reviewer, we identified only a few true grammatical errors, and we sincerely appreciate the insights provided by the reviewer, all of which have been addressed and corrected.**

Furthermore, the very important figure 1 is somewhat better than the previous figure - but still (due to poor execution), confusing and far from a professional sketch that the paper requires (and deserves).
**We have remade figure 1 and believe that this iteration is more intelligible and representative of the presented system.**

Finally, I don't know if this is due to being unexperienced or active ignorance or not taking the reviewers serious: (At least) in three cases in the answer to reviewers comments the authors write that they changed the text according to the reviewers suggestion - but actually did not do so:

1) they still write 'setpoint' instead of 'nominal value'

**We have certainly taken all comments by the reviewer seriously. Note that the first example was indeed changed in correspondence with the line number in the revised manuscript. However, we did not change 'setpoint' to 'nominal' throughout as it was not explicitly suggested. The native English language lead author and coauthors felt that 'setpoint' was a very suitable word choice in our case as it is commonly used. However, as this is a minor point and following the specific recommendation of the reviewer, we have now replaced 'setpoint' with 'nominal' throughout.**

2)  they still write 'valve-open percentage' instead of 'valve opening percentage'

**The reviewer is correct and we apologize for this overlook. We have changed it in the revised version.**

[Both are no major points, still they should not write that they changed it - and then don't...!]

3) In their responses to the 1st reviews, they advertise a figure on oxygen data from the mesocosms, which would have improved the paper, significantly. However, its missing in this version....

I don't know what to say...

**We believe this is a misunderstanding. To resolve this, we quote our first reply to the reviewer below**:

"Finally, the structure of the paper as well as the language are not always clear and precise (you may ask a native speaker to review the text) and – as no biological data are shown – the proof is missing that the system can create a biological effect (I don't really doubt this but it has to be shown).

**« We have gone through the manuscript with finer detail to ensure lucidity throughout. We respectfully disagree with the reviewer that there needs to be evidence of a biological effect. As per the suggestion of reviewer # 2, we have chosen to remove details about the biological component and focus more on the system itself. We agree with this suggestion as the aim of the manuscript is to show the functionality and assess the potential to manipulate environmental parameters, not to assess a biological response. We have, however, provided a figure below (see attachment PDF) showing the continuous measurement of oxygen during an incubation to provide a visual of the biologically induced response. Experimental results that utilized this system are Miller et al. (community-level effects, sbm: doi:10.2139/ssrn.4563719) and Lebrun et al. (organismal-level effects, sbm: 10.5194/egusphere-2023-1875). »**

**Perhaps the reviewer believed that our objective was to add this figure to the revised manuscript. However, we only provided the figure in the response to the reviewer's original comment for clarity.**

**Thus, we certainly acknowledge the reviewer's comment, as we did before, and provided evidence with our previous response. At that juncture, we choose to remove nearly all mention of the biological data (as suggested by reviewer # 2 during the first round of review), making the addition of an oxygen figure irrelevant—this was our reasoning for not including the figure in the manuscript. We did not choose to ignore this comment but found no room in the new version of the manuscript where such a figure would be relevant.**

Details (again, mixed minor and major):
L 72: '…will be affected when / while assemblages adjust…' _ **We have changed the wording and replaced "as" with "while".**

L 74: 'Methodological approaches to assesS and characterizE the responseS of organisms…' _ **This sentence has been rewritten as the changes suggested by the reviewer did not follow parallel structure with the rest of the sentence. We have incorporated the suggested changes by the reviewer in the revised sentence.**

LL 74-79: Sentence too long (and weird): Make a full-stop after the parentheses in L 77! **This sentence has been rewritten and split where suggested**.

L 109: write 'above' _instead of 'previously' _ **Done.**

L 112: 'In THE initial deployment…' _ **Done.**

L 143: replace 'shoal' _by 'reduce' _ **Done.**

L 144: '…in the Kongsfjorden environment…' _ **Changes done.**

L 144–146: Make it two sentences: '… _deployment of the SalTExPreS. It was placed on a concrete platform…' _ **We have split this sentence and adopted the suggestions.**

L 148: Replace 'represent' _by 'simulate' _ **Done.**

L 149: Replace 'examined' _by 'supervised' _ **Done.**

LL 149-150: Rephrase. It sounds like the communities can be found at 7m depth for 54 days, while you want to say that the experiment took 54 days… _ **We have rephrased this sentence to address the concern raised by the reviewer. We now state:**
**"The SalTExPreS was utilized to implement three treatment scenarios in a fractional-factorial design to simulate expected future conditions in Kongsfjorden for a 54-d experiment that supervised the productivity, survival, and growth response of mixed kelp communities surveyed at 7 m (maximum depth of collection)."**

L 152: '- the details': delete 'the' _ **Done.**

L 150–153: Move this sentence to the end of the paragraph! **This sentence has been moved to the end of the paragraph.**

L 157: '11m' _confusing: You want to simulate a -7m community but measure the fjord conditions at 11m??? **The 7 m depth was the maximum collection depth and the depth of comprehensive surveys describing community assemblages in the literature. The underwater observatory COSNYA site is located at a depth of 11 m. We did not have the ability to relocate the fixed COSNYA monitoring station. We have rewritten the sentence to:  "The chosen treatment and salinity perturbations were applied as offset values from *in-situ* fjord conditions, which were measured at an underwater observatory fixed at 11 m depth and captured the natural variability of the fjord system."**

L 166: 7m or 11m??? **This is 7 m as stated.**

L 171: Now its 10m??? Clarify! **The pump was situated at 10 m which was 1 m higher than the 11 m *in-situ* data measurements at the underwater observatory. This was done to prevent clogging from sediment which we had experienced before the experiment began. We have added the following in the revised manuscript:**
**"To prevent clogging from sediment, the pump was situated at a 10 m depth ensuring a safe height above sediment resuspension from the floor."**

L 172: '… _that was tapped into an underwater intake pipe AND that fed a header tank…' _ **Done.**

L 176: 'plumbed'…? What do you ant to say? **We do not understand the reviewer's question. Plumbed is the past participle of plumb and is the intended word.**

L 178 and L181: Dot missing after 'Fig' _ **Done.**

L 182: '…from the tap which IS fed by the…' _(guess this is still the case.) **Done.**

L 185: delete last part of the sentence (', where flow rates of 7 – _8 L…' _) as this is redundant to the previous information given. **This portion of the sentence has been deleted and rewritten as suggested.**

L 186: replace 'incubations' _by 'interruptions' _ **Done.**

L 188 write: '(3 treatments and 1 control, each with 3 replicates)' _ **Done as suggested.**

LL 191-192 write: 'Fiberglass insulation at the outside of each mesocosm reduced unintended changes in treatment water temperature.' _ **We have changed this sentence to reflect the suggestion by the reviewer.**

L 193: delete comma after 'warmed' _ **The Oxford comma has been deleted.**

L 196: no hyphen between flow and line_ **Done.**

LL 197-198: move '12 in total' _to the end of the sentence : (12 in total, Fig. 1)_**Done.**

LL 203–205: redundant and commonplace: rephrase! **We have rewritten these two sentences and removed any redundancy.**

L 205: delete 'all' _ **Done.**

L 206: delete comma after 'valves' _ **We disagree; the Oxford comma for this sentence improves clarity for the listed actions.**

LL 206–207 just write: '…logged every minute and displayed on the user interface (Fig. A3).' _ **We have revised the sentence accordingly.**

L 209 (and throughout the whole manuscript!): r_e_p_l_a_c_e__'s_e_t_p_o_i_n_t'_b_y_ _'n_o_m_i_n_a_l__v_a_l_u_e_' _ **"setpoint" has been replaced with " "nominal" or "nominal value" here and throughout**.

L 231: 1st mention of 'PLC': define here as 'Programmable Logic Controller (PLC)'!
**Done.**

L 231: '…informing ON proper communication…' _ **Done.**

L 233 (and elsewhere): r_e_p_l_a_c_e__'v_a_l_v_e_-o_p_e_n__p_e_r_c_e_n_t_a_g_e'__b_y__'v_a_l_v_e__o_p_e_n_i_n_g__p_e_r_c_e_n_t_a_g_e'__ **We have removed the stylistic compound modifier and changed "open to "opening." There were no other sentences where this needed to be done.**

L 237: % is no unit of concentration! Write: '.., O2 saturation (%),..' _ **We have corrected this error.**

L 247, 249 and elsewhere: replace 'control condition' _by 'control treatment' _or just 'control' _ **We have changed "control condition" to just "control" here and throughout.**

L 251: comma after 'period' _ **Due to the restructuring of the sentence as mentioned below, a comma is not needed here. See response to comment below.**

L 251 'across the 3 replicates', clarify: does this mean this is the average off the deviation over all 3 control replicates - or was the average in each replicate less than 0.3°C? **The reviewer's second interpretation is correct. We have adjusted the sentence to remove any ambiguity in the interpretation by the reader.**

L 251-254: replace 'was based on' _by 'was achieved by'. Furthermore: This sentence doesn't make sense at all: The ability to read data does not help to keep regulation quality high (low deviation from the nominal value). The fast response of the system to changes in the ferrybox data is the key player here... **We have changed "based on" to "achieved by" and replaced the**

word "read" with "interpret and respond to…" which we hope gives better clarity to the message being conveyed. This change also highlights the comment by the reviewer's reference to "fast response."

L 253-254: put last part of the sentence (after the comma) into parentheses. **Done.**

L 262 write: '…situated at 90 m depth in the fjord was used from…' _ **We have added the words "depth in the fjord" to the sentence as suggested.**

L 263-264: write: '…at 10 m depth was repaired.' _(motor malfunction info is irrelevant) **Done.**

L 275: replace 'deployment' _by e.g. 'experiment' _ **Done.**

L 281-282: delete: '…after the final incremental increase was programmed.' _ **Done.**

L 2282-284, Rephrase: I don't understand this sentence… _ **This sentence has been split for clarity.**

L 303-304: Please discuss: The ability to add warmed seawater also to the control system would allow for keeping the control stable even when using the 90m backup pump (in a future setup). **We have added a sentence following this one to reflect the statement made by the reviewer.**

L 311: replace 'inaugural' _by 'first' _ **We have replaced "inaugural" with "first."**

L 317: replace 'permits the' _by 'allows for the' _ **We have replaced "permits" with "allows for."**

L 325, write: 'Since the initial experiment, we implemented a number of changes to improve the performance of the system which have been realized during a second experiment in …' _ **We have changed this sentence to the one suggested by the reviewer.**

L 332-334, write: '…kept deviations in the 9 different mesocosms at < 0.5°C for 94% of the time (79% in the first experiment). **Done.**

L 334: 'During the first experiment…' **Done.**

L 335: 'largeR deviations' _ **Done.**

L 336: '…flow rates OF < 2 L min-1…' _ **Done.**

L 337: Simple software modifications_ **Done.**

L 340-341, just write: '…data were maintained solved most of the issues.' _ **We have changed this sentence to reflect the reviewer's suggestion.**

L 343, write: '…and clogging of the seawater inlet are issues that need to be addressed whenever…' _ **This sentence has been revised to reflect the latter portion of the suggested changes. The sentence now includes "…and clogging of the seawater intake ports." This is a slightly different suggestion than given by the reviewer but is more accurate given how the pump functions.**

L 345: How do you prevent clogging or remove the stuck material? Can the pumping direction be reversed? **The pump was 1 m above the seabed as explained in the revised manuscript (lines 172–173). It provides reasoning for the 10 m and 11 m difference between the pump depth and the COSYNA Observatory logging depth.**

L 346-347, write …' _independently regulate experimental conditions in a …' _ **Done.**

L 350: replace 'its' _by 'the system's' _ **Done.**
L 351: new sentence after '(e.g. tidally).' _

L 351: replace 'mimic a future scenario' _by 'mimic future scenarios' _ **Done.**

Figure 1:
- Sensor labels too small
- 'Control' _doesn't make sense as the violet/dark blue line also supplies the 'treatments'
- missleading: are the yellow marked valves 'pressure regulation valves' or 'pressure sensors' (as stated in the text below)? Or aren't the yellow marked things in the photos the same as in the sketch?
- Black line: where does the freshwater enter the system (and where starts the arrow that points at the hose in the photo)
- Overall: using some colorful lines as the actual pipes/tubes of the system - and other colorful lines as arrows that point at the fotos is very confusing and generally unacceptable for a scientific sketch.

I suggest that you use a number code (like in the foto - and write the same numbers next to the respective line or valve in the sketch.
- It is very important to be exact into each single detail in this figure!
**Figure 1 has been redrawn to address these comments.**

- L 539: that lead to all 12 3-way regulator valves **We have added "3-way" to the text.**

- L 544: photosynthetically without capital 'P' _ **Done.**

Figure 2:
- L 560-561: Not 'regulation': Never write what can be interpreted from a figure in the caption, but just what can be seen! Just write: 'Mean temperature offsets of all applied conditions. Blue: offset of the control from the FerryBox. Dark green, light green and yellow: offsets of the

treatments 1, 2 and 3 from the control, respectively.'  **We have removed "regulation of the" from the sentence and added two additional sentences as suggested by the reviewer.**

- L 562: end sentence after 'standard deviation'. (Short sentences are good!) **Done.**

Figure 3: 'mean' values: averaged across how many single datapoints? resp. across how much time???
**We have added the details asked for by the reviewer to the figure caption: values occur every minute spanning more than 60 d.**